# Quasiparticle disintegration in fermionic superfluids

Senne Van Loon[1,2⋆], Jacques Tempere[1,3] and Hadrien Kurkjian[1,4]

**1** TQC, Universiteit Antwerpen, Universiteitsplein 1, B-2610 Antwerpen, België
**2** School of Physics, Georgia Institute of Technology, Atlanta, Georgia 30332, USA
**3** Lyman Laboratory of Physics, Harvard University, Cambridge, Massachusetts 02138, USA
**4** Laboratoire de Physique Théorique, Université de Toulouse, CNRS, UPS, France

⋆ Senne.VanLoon@UAntwerpen.be

## Abstract

We study the fermionic quasiparticle spectrum in a zero-temperature superfluid Fermi gas, and in particular how it is modified by different disintegration processes. On top of the disintegration by emission of a collective boson ($1 \rightarrow 2$, subject of a previous study, PRL 124, 073404), we consider here disintegration events where three quasiparticles are emitted ($1 \rightarrow 3$). We show that both disintegration processes are described by a $t$-matrix self-energy (as well as some highly off-resonant vacuum processes), and we characterize the associated disintegration continua. At strong coupling, we show that the quasiparticle spectrum is heavily distorted near the $1 \rightarrow 3$ disintegration threshold. Near the dispersion minimum, where the quasiparticles remain well-defined, the main effect of the off-shell disintegration processes is to shift the location of the minimum by a value that corresponds to the Hartree shift in the BCS limit. With our approximation of the self-energy, the correction to the energy gap with respect to the mean-field result however remains small, in contrast with experimental measurements.

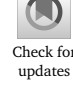

## 1   Introduction

Although very useful to characterize the state and dynamics of a many-body system [1,2] the concept of quasiparticles is quite often only an approximation of its spectral properties. This is particularly so in three-dimensional ergodic systems where quasiparticles are affected by several decay channels whose energy-momentum conservation constraint is met by exploiting the angular degrees of freedom [3]. The resonance of the Green's function at the eigenenergy of the quasiparticle is then broadened by a nonzero damping rate and diluted by a smaller spectral weight. The proximity of a decay threshold often distorts the quasiparticle dispersion relation and in some special cases it can entirely wash away the resonance [4]. These universal phenomena affect systems as diverse as normal [5,6] or superfluid Fermi liquids [7–9], superconductors [10], rotonic systems [11,12], or nuclear matter [13–15].

This article is devoted to fermionic pair condensates, which support two kinds of elementary excitations: the fermionic quasiparticles, describing unpaired fermions surrounded by the condensate of pairs, and the bosonic modes describing the collective motion of the pairs, such as sound waves in neutral gases [16–18], or plasma oscillations in charged systems [10,19]. In addition, a high-energy collective branch can exist slightly above the pair-breaking threshold [20–24]. Throughout this article, we restrict the word "quasiparticles" to the single-particle (fermionic) elementary excitations, and "collective modes" for the pair excitations. The properties of both kinds of elementary excitations are essential to understand transport [25] and dissipation [26] phenomena, or the weakly-excited dynamics of these fermionic condensates [27–29].

Neutral fermionic condensates are now routinely prepared using ultracold fermions [9, 30–34]. On this highly versatile platform, one can study the entire crossover from a BCS-like superfluid, to a Bose-Einstein condensate (BEC) of closely bound pairs by tuning the *s*-wave scattering length describing the interaction between fermionic atoms in different hyperfine states. By changing the magnetic field over a Feshbach resonance, it is even possible to reach the unitary regime where interactions are resonant. Using the set of techniques available to ultracold atomic experiments, such as Bragg [35–38] or rf-spectroscopy [23, 39–42], flat-bottom potentials [43,44] or interaction quenches, numerous experimental results have been accumulated on the collective mode spectrum [36–38], single-particle excitations [42,45], and the energy gap [38, 41], some of them being already beyond our state-of-the-art theoretical understanding.

At zero temperature, the mean-field BCS theory [46] describes non-interacting, undamped quasiparticles whose gap coincides with the order parameter $\Delta$. For collective modes, the RPA [19] predicts a similarly undamped phononic collective branch [16,17]. It is now established that beyond the mean-field approximation the ideal quasiparticle picture only works at low

energies. Away from their dispersion minimum, the quasiparticles can disintegrate [47–49], resulting in a nonzero damping. The gapped energy spectrum of the fermionic excitations makes sure that the main decay mechanism at low energies is the disintegration by emission of a phonon [50,51] (a sound wave of the Fermi gas, no to be confused with a lattice phonon in solid-state physics). This kind of interaction between fermionic quasiparticles and collective modes was considered in several studies, either from the point of view of collective modes [26, 50–52], or to study their effect on the quasiparticle properties [49, 53, 54]. Although the order parameter cannot be immediately related to the quasiparticle gap beyond BCS theory, we also note that there exists several calculations of this quantity [41, 55–57] beyond the mean-field BCS approximation.

In a previous study, we studied corrections to the quasiparticle dispersion relation focusing on the effect of the disintegration by emission of a collective mode ($1 \rightarrow 2$). Here we add the effect of disintegration into three quasiparticles ($1 \rightarrow 3$). Both disintegration processes are naturally described by the $t$-matrix self-energy [54, 58, 59]. In Sec. 2, we show that the $1 \rightarrow 2$ and $1 \rightarrow 3$ disintegrations are respectively associated with the poles and the branch cuts of the pair propagator, we derive the corresponding coupling amplitudes in the perturbative limit, and discuss the structure of the disintegration continua. We then study the full quasiparticle Green's function (Sec. 3), where the two processes interfere, and extract the corrected quasiparticle energy and lifetime. At strong coupling, we observe a strong distortion of the quasiparticle resonance around the $1 \rightarrow 3$ disintegration threshold $3\Delta$. Finally, we characterize the undamped low-energy region of the quasiparticle dispersion by measurable quantities such as the gap, effective mass and location of the dispersion minimum.

## 2 Single-particle propagator

We consider a two-component Fermi gas with neutral atoms of mass $m$, interacting with a short-range potential. For a dilute and ultracold gas the true interaction potential can be replaced by an effective attractive contact potential $g_0 \delta(\mathbf{r} - \mathbf{r}')$. The coupling constant $g_0$ should be renormalized [34]

$$\frac{1}{g_0} = \frac{m}{4\pi\hbar^2 a} - \frac{1}{V} \sum_{\mathbf{k}} \frac{m}{\hbar^2 \mathbf{k}^2}, \qquad (1)$$

to keep the pair propagator finite, where $a$ is the $s$-wave scattering length that determines the interaction regime.

Quite generally, when a many-body system supports quasiparticles, they appear as poles of the single-particle Green's function $G$. In the superfluid phase of a two-component Fermi gas, the single-particle Green's function is a two-by-two matrix with particle and hole Green's functions on the diagonal and anomalous Green's function on the codiagonal. In this situation, the poles of $G$ are given by [1]:

$$\det G^{-1}(\mathbf{k}, z_{\mathbf{k}}) = 0. \qquad (2)$$

Here $\mathbf{k}$ is the wave vector of the quasiparticle (we consider a homogeneous system for which $\mathbf{k}$ is a good quantum number) and $z_{\mathbf{k}}$ its eigenenergy, possibly taking complex values. To fully characterize the quasiparticle resonance in the Green's function, one should also introduce the matrix residue $Z_{\mathbf{k}}$ associated to the pole in $z_{\mathbf{k}}$, such that

$$G(\mathbf{k}, z) \underset{z \rightarrow z_{\mathbf{k}}}{\sim} \frac{Z_{\mathbf{k}}}{z - z_{\mathbf{k}}}. \qquad (3)$$

---

[1]We assume here that the matrix elements of $G^{-1}$ do not vanish individually, which is generally the case.

Note that the full Green's function $G(\mathbf{k}, z)$ contains in general more information than summarized by the quasiparticle spectrum.

The mean-field BCS theory provides a zeroth order approximation of the Green's function:

$$G^{(0)}(\mathbf{k}, z) = \frac{1}{z + \epsilon_\mathbf{k}} \begin{pmatrix} -V_\mathbf{k}^2 & U_\mathbf{k} V_\mathbf{k} \\ U_\mathbf{k} V_\mathbf{k} & -U_\mathbf{k}^2 \end{pmatrix} - \frac{1}{z - \epsilon_\mathbf{k}} \begin{pmatrix} U_\mathbf{k}^2 & U_\mathbf{k} V_\mathbf{k} \\ U_\mathbf{k} V_\mathbf{k} & V_\mathbf{k}^2 \end{pmatrix}, \tag{4}$$

$$(G^{(0)}(\mathbf{k}, z))^{-1} = \begin{pmatrix} -z + \xi_\mathbf{k} & \Delta \\ \Delta & -z - \xi_\mathbf{k} \end{pmatrix}. \tag{5}$$

In this approximation, the quasiparticle energy is

$$\epsilon_\mathbf{k} = \sqrt{\xi_\mathbf{k}^2 + \Delta^2}, \tag{6}$$

with $\xi_\mathbf{k} = \hbar^2 k^2 / 2m - \mu$ the energy of the free fermions, $\mu$ the chemical potential, and $\Delta$ the order parameter, acting also as a gap of the dispersion relation $\epsilon_\mathbf{k}$. Note that the minimum of the dispersion relation is reached at the wave vector

$$\hbar k_m^{(0)} = \sqrt{2m\mu}. \tag{7}$$

The Bogoliubov coefficients

$$U_\mathbf{k} = \sqrt{\frac{1}{2}\left(1 + \frac{\xi_\mathbf{k}}{\epsilon_\mathbf{k}}\right)}, \quad V_\mathbf{k} = \sqrt{\frac{1}{2}\left(1 - \frac{\xi_\mathbf{k}}{\epsilon_\mathbf{k}}\right)}, \tag{8}$$

can be interpreted as the weights of the quasiparticle on the particle and hole channels: when $U_\mathbf{k}$ increases the quasiparticle resembles more a particle excitation, whereas when $V_\mathbf{k}$ increases it resembles a hole excitation. These coefficients allow us to diagonalize the Green's function:

$$B_\mathbf{k} \left(G^{(0)}\right)^{-1} B_\mathbf{k}^\dagger = \begin{pmatrix} -z + \epsilon_\mathbf{k} & 0 \\ 0 & -z - \epsilon_\mathbf{k} \end{pmatrix}, \qquad \text{with} \quad B_\mathbf{k} = \begin{pmatrix} U_\mathbf{k} & V_\mathbf{k} \\ -V_\mathbf{k} & U_\mathbf{k} \end{pmatrix}. \tag{9}$$

The fact that mean-field theory describes quasiparticles with purely real eigenenergies is a serious limitation, since the quasiparticle lifetime plays an important role in many dissipation mechanisms. This is one of the motivations to introduce a correction to the Green's function in the form of a self-energy:

$$G^{-1}(\mathbf{k}, i\hbar\omega_n) = (G^{(0)}(\mathbf{k}, i\hbar\omega_n))^{-1} - \Sigma(\mathbf{k}, i\hbar\omega_n), \tag{10}$$

where $\Sigma$ is defined a priori only for imaginary fermionic Matsubara frequencies $i\hbar\omega_n = i(2n+1)\pi k_\mathrm{B} T$, with $n \in \mathbb{Z}$. In this work, we consider the so-called $t$-matrix self-energy [54, 58, 59]:

$$\Sigma_{ss'}(\mathbf{k}, i\hbar\omega_n) = -ss' \frac{1}{\beta V} \sum_{\mathbf{q}, m} G^{(0)}_{s', s}(\mathbf{q} - \mathbf{k}, i\hbar\nu_m - i\hbar\omega_n) \Gamma_{ss'}(\mathbf{q}, i\hbar\nu_m), \tag{11}$$

with $s, s' \in \{+, -\}$.

The self-energy of Eq. (11) can be studied with various degrees of self-consistency [59]. Here, we choose to use a completely non-self-consistent approach that allows for an analytic study of the different processes contributing to the quasiparticle spectrum. Then, the ladder-resummed pair propagator $\Gamma(\mathbf{q}, i\hbar\nu_m)$, with $\nu_m = 2m\pi k_\mathrm{B} T / \hbar$ bosonic Matsubara frequencies, is given in terms of the mean-field single-particle propagators

$$\Gamma_{ss'}(\mathbf{q}, i\hbar\nu_m) = \left(-\frac{1}{g_0}\delta_{ss'} + N_{ss'}(\mathbf{q}, i\hbar\nu_m)\right)^{-1}, \tag{12}$$

$$N_{ss'}(\mathbf{q}, i\hbar\nu_m) = \frac{1}{\beta V} \sum_{\mathbf{k}, n} G^{(0)}_{ss'}(\mathbf{k} + \frac{\mathbf{q}}{2}, i\hbar\omega_n + \frac{i\hbar\nu_m}{2}) G^{(0)}_{-s', -s}(\mathbf{k} - \frac{\mathbf{q}}{2}, i\hbar\omega_n - \frac{i\hbar\nu_m}{2}), \tag{13}$$

where $N$ is the bare pair propagator. We note that $\Gamma$ is an even function of its energy argument, $\Gamma(\mathbf{q}, -z_q) = \Gamma(\mathbf{q}, z_q)$, which ensures that its spectrum is symmetric about the imaginary axis.

To calculate the pair propagator, we make use of the mean-field equation for the order parameter

$$\Delta = -\frac{g_0}{\beta V} \sum_{\mathbf{k},n} G_{+-}^{(0)}(\mathbf{k}, i\hbar\omega_n), \qquad (14)$$

to replace $g_0$ in Eq. (12). In this way, the equations explicitly only depend on the natural parameters $\Delta$ and $\mu$, in favor of the $s$-wave scattering length $a$. The interaction regime is then completely determined by fixing the dimensionless coefficient $\mu/\Delta$. To relate the parameters of the theory $\Delta$ and $\mu$ to the experimentally more relevant interaction parameter $1/k_F a$ and the Fermi wavevector $k_F$ (effectively fixing the density through $n = k_F^3/3\pi^2$), the order parameter equation (14) has to be solved together with the number equation, which in the mean-field approximation is given by

$$n = \frac{1}{V} \sum_{\mathbf{k}} \left[ 1 - \frac{\xi_{\mathbf{k}}}{\epsilon_{\mathbf{k}}} \right]. \qquad (15)$$

One can replace this number equation to include beyond-mean-field corrections, which can lead to more quantitatively correct results. To avoid making a choice for the approximation of the number equation, we present our main results in terms of the coefficient $\mu/\Delta$. Whenever we relate this to the coupling parameter $1/k_F a$, we use the mean-field number equation, as it allows for an analytic solution [16].

At zero temperature, the Matsubara sum in the self-energy [Eq. (11)] becomes an integral over the imaginary frequencies $\sum_m \to \int_{-i\infty}^{i\infty} \frac{dz_q}{2\pi i}$. Replacing $G^{(0)}$ by its BCS form (4), we cast the self-energy in the form:

$$\Sigma(\mathbf{k}, i\hbar\omega_n) = \frac{1}{V} \sum_{\mathbf{q}} \int_{-i\infty}^{i\infty} \frac{dz_q}{2\pi i} \left[ \frac{1}{z_q - i\hbar\omega_n + \epsilon_{\mathbf{k}-\mathbf{q}}} \begin{pmatrix} V_{\mathbf{k}-\mathbf{q}}^2 \Gamma_{++}(\mathbf{q}, z_q) & U_{\mathbf{k}-\mathbf{q}} V_{\mathbf{k}-\mathbf{q}} \Gamma_{+-}(\mathbf{q}, z_q) \\ U_{\mathbf{k}-\mathbf{q}} V_{\mathbf{k}-\mathbf{q}} \Gamma_{-+}(\mathbf{q}, z_q) & U_{\mathbf{k}-\mathbf{q}}^2 \Gamma_{--}(\mathbf{q}, z_q) \end{pmatrix} \right.$$

$$\left. - \frac{1}{z_q - i\hbar\omega_n - \epsilon_{\mathbf{k}-\mathbf{q}}} \begin{pmatrix} -U_{\mathbf{k}-\mathbf{q}}^2 \Gamma_{++}(\mathbf{q}, z_q) & U_{\mathbf{k}-\mathbf{q}} V_{\mathbf{k}-\mathbf{q}} \Gamma_{+-}(\mathbf{q}, z_q) \\ U_{\mathbf{k}-\mathbf{q}} V_{\mathbf{k}-\mathbf{q}} \Gamma_{-+}(\mathbf{q}, z_q) & -V_{\mathbf{k}-\mathbf{q}}^2 \Gamma_{--}(\mathbf{q}, z_q) \end{pmatrix} \right]. \qquad (16)$$

While computing the self-energy by integrating over $z_q$ on the imaginary axis is a frequent and perfectly conceivable strategy (including numerically), here we prefer to deform the integration contour towards the real axis, as shown in Figure 1. This strategy allows us to isolate the contribution of each pole or branch cut of the pair propagator to the self-energy, and to interpret them separately in terms of elementary decay processes affecting the quasiparticle. The straightforward analytic continuation of $\Gamma$ to complex frequencies ($i\hbar\nu_m \to z_q$) reveals its analytic structure on the real axis (in black on Figure 1). Firstly, there are two real poles in $z_q = \pm\hbar\omega_{\mathbf{q}}$, representing the phononic collective branch of a neutral Fermi gas [19]. Secondly, there are two gapped branch cuts at energies where it is possible to break up pairs. Opposite in energy, those branch cuts are bound by the pair-breaking threshold $\pm\epsilon_c(\mathbf{q})$ with

$$\epsilon_c(\mathbf{q}) = \min_{\mathbf{k}} [\epsilon_{\mathbf{k}-\mathbf{q}/2} + \epsilon_{\mathbf{k}+\mathbf{q}/2}] = \begin{cases} 2\Delta, & \text{if } q \leq 2k_m^{(0)} \\ 2\epsilon_{q/2}, & \text{if } q > 2k_m^{(0)} \end{cases}. \qquad (17)$$

On top of the analytic structure of $\Gamma$, the free single-particle propagator $G^{(0)}$ gives rise to two additional complex poles, in $z_q = i\hbar\omega_n - \epsilon_{\mathbf{k}-\mathbf{q}}$ [for the first term in Eq. (16)] and $i\hbar\omega_n + \epsilon_{\mathbf{k}-\mathbf{q}}$ (for the second term). To evade those poles and focus on the singularities of the

pair propagator, we choose different contours for these two terms: for the first term of Eq. (16) we deform the contour to the positive real axis, (see contour $\mathcal{C}_1$ in Fig. 1), while for the second term the contour is deformed to the negative real axis (contour $\mathcal{C}_2$ in Fig. 1).

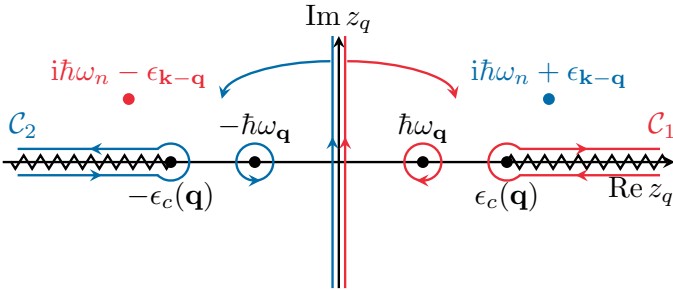

Figure 1: The analytic structure of the integrand of the self-energy $\Sigma(\mathbf{k}, i\hbar\omega_n)$. The Matsubara integral over the imaginary axis can be performed by analytically continuing the Matsubara frequencies and computing a contour integral. The contours are chosen in such a way to avoid the poles of the single particle propagator, resulting in the two keyhole contours $\mathcal{C}_1$ and $\mathcal{C}_2$ used respectively in the first and second term of Eq. (16).

This procedure allows us to identify two physically distinct contributions to the self-energy. The residues of the poles of the pair propagator will give rise to a coupling between the fermionic quasiparticles and the collective modes, such as the emission of a phonon by the quasiparticle, a process which has been studied in-depth in Ref. [48]. Conversely, the branch cut contributions are related to four-fermion processes, such as the decay of a quasiparticle into three, as we explain below.

We expect that these disintegration processes, described by the self-energy of Eq. (11), are the main decay channels of the quasiparticles at low energy. Unfortunately, there is no small parameter that can be used to estimate the importance of higher-order processes. Such processes could include the emission of two or more bosons by a quasiparticle, or the disintegration of a quasiparticle into five or more. The former should lead to a small correction at low temperature [60], while the latter only become resonant above $5\Delta$, and should thus be suppressed at low energies.

## 2.1 Contribution of the pole

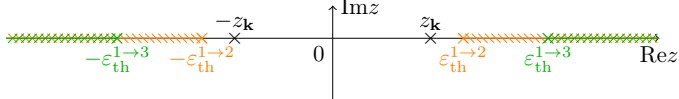

Figure 2: Analytic structure of the self-energy $\Sigma(\mathbf{k}, z)$ on the real axis. The two branch cuts on the positive real axis represent the real quasiparticle decay processes (orange stripes: the process $\gamma \to \gamma + b$ associated to the bosonic continuum $[\epsilon_{th}^{1\to2}, +\infty[$, green stripes: the process $\gamma \to \gamma + \gamma + \gamma$ associated to the fermionic continuum $[\epsilon_{th}^{1\to3}, +\infty[$, in terms the fermionic quasiparticle $\gamma$ and a bosonic collective mode $b$). Below those thresholds, a real solution $z_{\mathbf{k}}$ of (2) may exists depending on the value of $\mathbf{k}$. The two branches on the negative real axis represent the far-off-shell virtual processes (orange stripes: the process $\emptyset \to \gamma+\gamma+b$, green stripes: the process $\emptyset \to \gamma+\gamma+\gamma+\gamma$).

The residue of the $z_q$-integral in the poles of $\Gamma$ is given by

$$\Sigma^{\mathrm{p}}(\mathbf{k}, z) = \frac{1}{V} \sum_{\mathbf{q}} \frac{\hbar}{\partial_\omega \det \Gamma^{-1}(\mathbf{q}, \omega_{\mathbf{q}})} \Biggl[ \tag{18}$$

$$\frac{1}{z - \hbar\omega_{\mathbf{q}} - \epsilon_{\mathbf{k}-\mathbf{q}}} \begin{pmatrix} V_{\mathbf{k}-\mathbf{q}}^2 \check{N}_{--}(\mathbf{q}, \omega_{\mathbf{q}}) & -U_{\mathbf{k}-\mathbf{q}}V_{\mathbf{k}-\mathbf{q}}N_{+-}(\mathbf{q}, \omega_{\mathbf{q}}) \\ -U_{\mathbf{k}-\mathbf{q}}V_{\mathbf{k}-\mathbf{q}}N_{+-}(\mathbf{q}, \omega_{\mathbf{q}}) & U_{\mathbf{k}-\mathbf{q}}^2 \check{N}_{++}(\mathbf{q}, \omega_{\mathbf{q}}) \end{pmatrix}$$

$$- \frac{1}{-z - \hbar\omega_{\mathbf{q}} - \epsilon_{\mathbf{k}-\mathbf{q}}} \begin{pmatrix} U_{\mathbf{k}-\mathbf{q}}^2 \check{N}_{++}(\mathbf{q}, \omega_{\mathbf{q}}) & U_{\mathbf{k}-\mathbf{q}}V_{\mathbf{k}-\mathbf{q}}N_{+-}(\mathbf{q}, \omega_{\mathbf{q}}) \\ U_{\mathbf{k}-\mathbf{q}}V_{\mathbf{k}-\mathbf{q}}N_{-+}(\mathbf{q}, \omega_{\mathbf{q}}) & V_{\mathbf{k}-\mathbf{q}}^2 \check{N}_{--}(\mathbf{q}, \omega_{\mathbf{q}}) \end{pmatrix} \Biggr],$$

where we have defined $\check{N}_{\pm\pm} = N_{\pm\pm} - 1/g_0$. The absence of singularities of $\Sigma$ outside the real axis allowed us to analytically continue it in a natural way from Matsubara to complex frequencies $i\hbar\omega_n \to z$. Looking at the poles of the integrand as a function of the wave vector $\mathbf{q}$, we remark that the first term of $\Sigma^{\mathrm{p}}$ in Eq. (18) has a branch cut on the positive real axis for $z \in \{\hbar\omega_{\mathbf{q}} + \epsilon_{\mathbf{k}-\mathbf{q}} \text{ for } \mathbf{q} \in \mathbb{R}^3\}$, while the second term has a symmetric branch cut on the negative real axis (see the orange stripes on Fig. 2). Restricting to $\mathrm{Re}\, z > 0$ without loss of generality, we relate these branch cuts to two elementary processes contributing to the correction of the single-particle Green's function: the term with $z - \hbar\omega_{\mathbf{q}} - \epsilon_{\mathbf{k}-\mathbf{q}}$ in the denominator describes the emission of a collective excitation $b$ by the quasiparticle ($\gamma \to \gamma + b$, in short $1 \to 2$), while the term with $-z - \hbar\omega_{\mathbf{q}} - \epsilon_{\mathbf{k}-\mathbf{q}}$ depicts the spontaneous emission out of vacuum of two quasiparticles and a collective excitation ($\emptyset \to \gamma + \gamma + b$, in short $0 \to 3$), as shown in figures 3a and 3b. The emission process can be resonant when the quasiparticle energy is larger than the threshold value

$$\epsilon_{\mathrm{th}}^{1\to2}(\mathbf{k}) = \min_{\mathbf{q}}[\epsilon_{\mathbf{k}-\mathbf{q}} + \hbar\omega_{\mathbf{q}}], \tag{19}$$

such that an undamped quasiparticle can exist only at energies $z_{\mathbf{k}} < \epsilon_{\mathrm{th}}^{1\to2}$. Conversely, the simultaneous emission out of vacuum ($0 \to 3$) is a far-off-shell process acting on the quasiparticle only through an energy shift.

## 2.2 Contribution of the branch cut

For the part of the contour along the branch cut, one obtains the following self-energy

$$\Sigma^{\mathrm{bc}}(\mathbf{k}, z) = \frac{1}{V} \sum_{\mathbf{q}} \int_{\epsilon_c(\mathbf{q})}^{+\infty} \mathrm{d}z_q \Biggl[ -\frac{1}{z_q - z + \epsilon_{\mathbf{k}-\mathbf{q}}} \begin{pmatrix} V_{\mathbf{k}-\mathbf{q}}^2 \rho_{++}(\mathbf{q}, z_q) & U_{\mathbf{k}-\mathbf{q}}V_{\mathbf{k}-\mathbf{q}}\rho_{+-}(\mathbf{q}, z_q) \\ U_{\mathbf{k}-\mathbf{q}}V_{\mathbf{k}-\mathbf{q}}\rho_{-+}(\mathbf{q}, z_q) & U_{\mathbf{k}-\mathbf{q}}^2 \rho_{--}(\mathbf{q}, z_q) \end{pmatrix}$$

$$- \frac{1}{-z_q - z - \epsilon_{\mathbf{k}-\mathbf{q}}} \begin{pmatrix} U_{\mathbf{k}-\mathbf{q}}^2 \rho_{--}(\mathbf{q}, z_q) & -U_{\mathbf{k}-\mathbf{q}}V_{\mathbf{k}-\mathbf{q}}\rho_{+-}(\mathbf{q}, z_q) \\ -U_{\mathbf{k}-\mathbf{q}}V_{\mathbf{k}-\mathbf{q}}\rho_{-+}(\mathbf{q}, z_q) & V_{\mathbf{k}-\mathbf{q}}^2 \rho_{++}(\mathbf{q}, z_q) \end{pmatrix} \Biggr], \tag{20}$$

where we have introduced the spectral density of the pair propagator

$$\rho_{ss'}(\mathbf{q}, z_q) = \frac{\Gamma_{ss'}(\mathbf{q}, z_q - i0^+) - \Gamma_{ss'}(\mathbf{q}, z_q + i0^+)}{2\pi i}. \tag{21}$$

This time, the first line of Eq. (20) has a branch cut for $z$ larger than the fermionic disintegration threshold $\epsilon_{\mathrm{th}}^{1\to3} = \min_{\mathbf{q}} \min_{\omega_q \geq \epsilon_c(\mathbf{q})}[\omega_q + \epsilon_{\mathbf{k}-\mathbf{q}}]$. This branch cut is shown as green stripes on the positive real axis of Fig. 2. The symmetric branch cut on the negative real axis stems from the second line of Eq. (20) and describes a far-off-shell process. Using the definition (17) of the pair-breaking threshold $\epsilon_c(\mathbf{q})$ one can rewrite $\epsilon_{\mathrm{th}}^{1\to3}$ in a form which suggests its physical origin:

$$\epsilon_{\mathrm{th}}^{1\to3}(\mathbf{k}) = \min_{\mathbf{k}_1, \mathbf{k}_2}[\epsilon_{\mathbf{k}_1} + \epsilon_{\mathbf{k}_2} + \epsilon_{\mathbf{k}-\mathbf{k}_1-\mathbf{k}_2}] \geq 3\Delta. \tag{22}$$

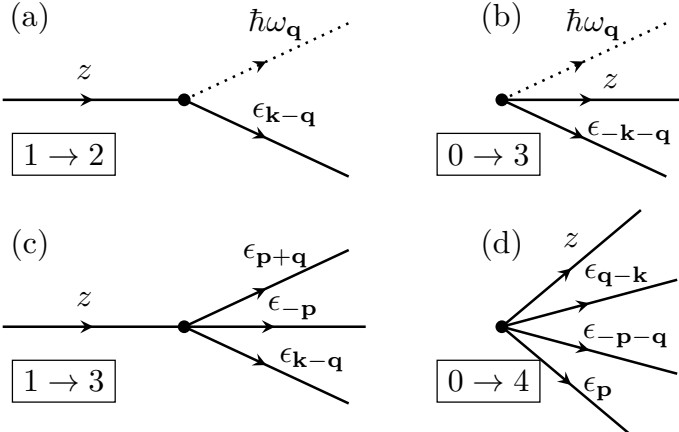

Figure 3: Different processes contributing to the fermionic self-energy. The full lines depict quasiparticles, while the dotted lines are collective modes. Both (a) and (b) result from the residue of the poles of the self-energy integrand of Eq. (16), and describe the emission of a collective mode by a quasiparticle and the spontaneous emission of two quasiparticles and a collective mode from the vacuum. The diagrams in (c) and (d) describe four-fermion processes, coming from the contribution of the branch cut. Only in the weak-coupling limit can an explicit coupling amplitude be written down for these processes.

The associated resonance condition $z_{\mathbf{k}} = \epsilon_{\mathbf{k}_1} + \epsilon_{\mathbf{k}_2} + \epsilon_{\mathbf{k}-\mathbf{k}_1-\mathbf{k}_2}$ indicates that the first line of $\Sigma^{\mathrm{bc}}$ describes the decay of the quasiparticle of momentum $\mathbf{k}$ into three other quasiparticles of momenta $\mathbf{k}_1$, $\mathbf{k}_2$ and $\mathbf{k}-\mathbf{k}_1-\mathbf{k}_2$ ($\gamma \to \gamma+\gamma+\gamma$, in short $1 \to 3$, see the diagram on Fig. 3c). The second line of $\Sigma^{\mathrm{bc}}$ then corresponds to the far-off-shell process where 4 quasiparticles appear out of vaccum ($\emptyset \to \gamma+\gamma+\gamma+\gamma$, in short $0 \to 4$, see diagram 3d). This term will not contribute to the imaginary part of the self-energy, but will nevertheless result in an energy shift of the quasiparticles.

## 2.3 Coupling amplitudes

To confirm our intuition that $\Sigma^{\mathrm{bc}}$ describes 4-fermion processes, we now express the corrected Green's function in terms of the coupling amplitudes associated to these processes. Due to the non-perturbative nature of the ladder-resummed pair propagator, this is not possible in the general case, but only in the perturbative limit. In this limit, we replace $g_0$ by $g = 4\pi\hbar^2 a/m$ in the pair propagator [Eq. (12)] and expand for $g \to 0$:

$$\Gamma_{ss'} = -g^2 \left( \frac{\delta_{ss'}}{g} + \frac{\delta_{ss'}}{V} \sum_{\mathbf{k}} \frac{m}{\hbar^2 \mathbf{k}^2} + N_{ss'} \right) + \mathcal{O}(g^3), \tag{23}$$

such that the spectral density $\rho$ takes the following form

$$\rho(\mathbf{q}, z_q) \xrightarrow{g\to 0} -\frac{g^2}{V} \sum_{\mathbf{p}} \left[ \begin{pmatrix} U_{\mathbf{p}_+}^2 U_{\mathbf{p}_-}^2 & -U_{\mathbf{p}_+} V_{\mathbf{p}_+} U_{\mathbf{p}_-} V_{\mathbf{p}_-} \\ -U_{\mathbf{p}_+} V_{\mathbf{p}_+} U_{\mathbf{p}_-} V_{\mathbf{p}_-} & V_{\mathbf{p}_+}^2 V_{\mathbf{p}_-}^2 \end{pmatrix} \delta(z_q - \epsilon_{\mathbf{p}_+} - \epsilon_{\mathbf{p}_-}) \right. \tag{24}$$
$$\left. - \begin{pmatrix} V_{\mathbf{p}_+}^2 V_{\mathbf{p}_-}^2 & -U_{\mathbf{p}_+} V_{\mathbf{p}_+} U_{\mathbf{p}_-} V_{\mathbf{p}_-} \\ -U_{\mathbf{p}_+} V_{\mathbf{p}_+} U_{\mathbf{p}_-} V_{\mathbf{p}_-} & U_{\mathbf{p}_+}^2 U_{\mathbf{p}_-}^2 \end{pmatrix} \delta(z_q + \epsilon_{\mathbf{p}_+} + \epsilon_{\mathbf{p}_-}) \right],$$

where we have introduced the short-hand notation $\mathbf{p}_\pm = \mathbf{p} \pm \mathbf{q}/2$. The Dirac delta functions in the spectral density can be used to perform the integral over $z_q$ in the self-energy $\Sigma^{\mathrm{bc}}$ of

Eq. (20), which exposes the elementary processes involving 4 fermionic quasiparticles:

$$\Sigma^{bc}(\mathbf{k},z) \xrightarrow{g \to 0} \frac{g^2}{V^2} \sum_{\mathbf{k}_1,\mathbf{k}_2,\mathbf{k}_3} \left[ \frac{\delta_{\mathbf{k}_1+\mathbf{k}_2+\mathbf{k}_3,\mathbf{k}}}{z - \epsilon_{\mathbf{k}_1} - \epsilon_{\mathbf{k}_2} - \epsilon_{\mathbf{k}_3}} \begin{pmatrix} -V_{\mathbf{k}_1}^2 U_{\mathbf{k}_2}^2 U_{\mathbf{k}_3}^2 & U_{\mathbf{k}_1} V_{\mathbf{k}_1} U_{\mathbf{k}_2} V_{\mathbf{k}_2} U_{\mathbf{k}_3} V_{\mathbf{k}_3} \\ U_{\mathbf{k}_1} V_{\mathbf{k}_1} U_{\mathbf{k}_2} V_{\mathbf{k}_2} U_{\mathbf{k}_3} V_{\mathbf{k}_3} & -U_{\mathbf{k}_1}^2 V_{\mathbf{k}_2}^2 V_{\mathbf{k}_3}^2 \end{pmatrix} \right.$$

$$\left. + \frac{\delta_{\mathbf{k}_1+\mathbf{k}_2+\mathbf{k}_3,-\mathbf{k}}}{-z - \epsilon_{\mathbf{k}_1} - \epsilon_{\mathbf{k}_2} - \epsilon_{\mathbf{k}_3}} \begin{pmatrix} U_{\mathbf{k}_1}^2 V_{\mathbf{k}_2}^2 V_{\mathbf{k}_3}^2 & U_{\mathbf{k}_1} V_{\mathbf{k}_1} U_{\mathbf{k}_2} V_{\mathbf{k}_2} U_{\mathbf{k}_3} V_{\mathbf{k}_3} \\ U_{\mathbf{k}_1} V_{\mathbf{k}_1} U_{\mathbf{k}_2} V_{\mathbf{k}_2} U_{\mathbf{k}_3} V_{\mathbf{k}_3} & V_{\mathbf{k}_1}^2 U_{\mathbf{k}_2}^2 U_{\mathbf{k}_3}^2 \end{pmatrix} \right]. \tag{25}$$

Finally, we rewrite the Green's function in the quasiparticle basis $\tilde{G}^{-1}(\mathbf{k},z) = B_{\mathbf{k}} G^{-1}(\mathbf{k},z) B_{\mathbf{k}}^\dagger$. In the quasiparticle-quasiparticle channel, we obtain

$$\tilde{G}_{++}^{-1}(\mathbf{k},z) \xrightarrow{g \to 0} -z + \epsilon_{\mathbf{k}} + \delta\epsilon(\mathbf{k},z), \tag{26}$$

with a complex energy shift $\delta\epsilon(\mathbf{k},z)$ that combines all the processes of Fig. 3:

$$\delta\epsilon(\mathbf{k},z) \equiv \frac{1}{V} \sum_{\mathbf{q}} \left[ \frac{\mathcal{A}_{\mathbf{k}-\mathbf{q},\mathbf{q}}^2}{z - \epsilon_{\mathbf{k}-\mathbf{q}} - \hbar\omega_{\mathbf{q}}} - \frac{\mathcal{B}_{\mathbf{k},\mathbf{q}}^2}{-z - \epsilon_{\mathbf{k}+\mathbf{q}} - \hbar\omega_{\mathbf{q}}} \right]$$

$$+ \frac{1}{V^2} \sum_{\mathbf{k}_1,\mathbf{k}_2} \left[ \frac{(\mathcal{A}_{\mathbf{k};\mathbf{k}_1,\mathbf{k}_2}^{1 \to 3})^2}{z - \epsilon_{\mathbf{k}_1} - \epsilon_{\mathbf{k}_2} - \epsilon_{\mathbf{k}-\mathbf{k}_1-\mathbf{k}_2}} - \frac{(\mathcal{A}_{\mathbf{k};\mathbf{k}_1,\mathbf{k}_2}^{4 \to 0})^2}{-z - \epsilon_{\mathbf{k}_1} - \epsilon_{\mathbf{k}_2} - \epsilon_{-\mathbf{k}-\mathbf{k}_1-\mathbf{k}_2}} \right]. \tag{27}$$

The coupling amplitudes $\mathcal{A}_{\mathbf{k}-\mathbf{q},\mathbf{q}}$ of the emission process $1 \to 2$ and $\mathcal{B}_{\mathbf{k},\mathbf{q}}$ of the vacuum process $0 \to 3$ are described by $\Sigma^p$ and discussed in detail in Ref. [48]. Here, we add the contribution of the fermionic disintegration process $1 \to 3$ and vacuum process $0 \to 4$ from $\Sigma^{bc}$. Their respective coupling amplitudes are given by

$$\mathcal{A}_{\mathbf{k};\mathbf{k}_1,\mathbf{k}_2}^{1 \to 3} = g(U_{\mathbf{k}} U_{\mathbf{k}_1} U_{\mathbf{k}_2} V_{\mathbf{k}-\mathbf{k}_1-\mathbf{k}_2} - V_{\mathbf{k}} V_{\mathbf{k}_1} V_{\mathbf{k}_2} U_{\mathbf{k}-\mathbf{k}_1-\mathbf{k}_2}), \tag{28}$$

$$\mathcal{A}_{\mathbf{k},\mathbf{k}_1,\mathbf{k}_2}^{4 \to 0} = g(U_{\mathbf{k}} U_{\mathbf{k}_1} V_{\mathbf{k}_2} V_{-\mathbf{k}-\mathbf{k}_1-\mathbf{k}_2} + V_{\mathbf{k}} V_{\mathbf{k}_1} U_{\mathbf{k}_2} U_{-\mathbf{k}-\mathbf{k}_1-\mathbf{k}_2}). \tag{29}$$

Note that the vacuum terms $0 \to 3$ and $0 \to 4$ appear with a minus sign in Eq. (27) because they occur in the ground state of the system $|\psi_0\rangle$ and not in the state containing one quasiparticle $\hat{\gamma}_{\mathbf{k}}^\dagger |\psi_0\rangle$.

The amplitudes (28)–(29) were derived in Ref. [54] [see Eqs. (B7–9) therein] by expressing the interaction Hamiltonian $\hat{H}_{\text{int}} = g \int d^3r \, \hat{\psi}_\uparrow^\dagger(\mathbf{r}) \hat{\psi}_\downarrow^\dagger(\mathbf{r}) \hat{\psi}_\downarrow(\mathbf{r}) \hat{\psi}_\uparrow(\mathbf{r})$ in terms of the quasiparticle creation-annihilation operators. In this weak-coupling limit the branch cut contribution to $\Sigma$ thus describes the damping and energy-shift due to the 4-fermion processes treated to second order in perturbation theory.

For completeness, we also give the Green's function in the quasiparticle-quasihole channel:

$$\tilde{G}_{+-}^{-1}(\mathbf{k},z) \xrightarrow{g \to 0} -\frac{1}{V} \sum_{\mathbf{q}} \left[ \frac{\mathcal{A}_{\mathbf{k}-\mathbf{q},\mathbf{q}} \mathcal{B}_{\mathbf{k}-\mathbf{q},\mathbf{q}}}{z - \epsilon_{\mathbf{k}-\mathbf{q}} - \hbar\omega_{\mathbf{q}}} + \frac{\mathcal{A}_{\mathbf{k},\mathbf{q}} \mathcal{B}_{\mathbf{k},\mathbf{q}}}{-z - \epsilon_{\mathbf{k}+\mathbf{q}} - \hbar\omega_{\mathbf{q}}} \right]$$

$$- \frac{1}{V^2} \sum_{\mathbf{k}_1,\mathbf{k}_2} \left[ \frac{\mathcal{A}_{\mathbf{k},\mathbf{k}_1,\mathbf{k}_2}^{1 \to 3} \mathcal{A}_{\mathbf{k},\mathbf{k}_1,\mathbf{k}_2}^{4 \to 0}}{z - \epsilon_{\mathbf{k}_1} - \epsilon_{\mathbf{k}_2} - \epsilon_{\mathbf{k}_3}} + \frac{\mathcal{A}_{\mathbf{k},\mathbf{k}_1,\mathbf{k}_2}^{1 \to 3} \mathcal{A}_{\mathbf{k},\mathbf{k}_1,\mathbf{k}_2}^{4 \to 0}}{-z - \epsilon_{\mathbf{k}_1} - \epsilon_{\mathbf{k}_2} - \epsilon_{\mathbf{k}_3}} \right]. \tag{30}$$

This non-zero off-diagonal matrix element shows that the quasiparticle described by $\tilde{G}$ is a mix of the original BCS quasiparticle and quasihole (or in other word that its weight on the particle and hole channels differ from the $U_{\mathbf{k}}$ and $V_{\mathbf{k}}$ prescribed by BCS theory). We note that $\tilde{G}_{+-}^{-1}$ is no longer small outside the limit $g \to 0$, such that ($i$) the quasiparticle energy $z_{\mathbf{k}}$ is no

longer given by $\epsilon_{\mathbf{k}} + \delta\epsilon(\mathbf{k}, z)$, and $(ii)$ the contribution of $\Sigma^{\mathrm{p}}$ and $\Sigma^{\mathrm{bc}}$ to $z_{\mathbf{k}} - \epsilon_{\mathbf{k}}$ can no longer be disentangled. This is the case even in the BCS limit as shown by our numerical results in the next section. As such, all the numerical results we present in Sec. 3 make use of the full self-energy of Eq. (16), valid for all couplings, and not the perturbative expressions given in this section [2].

## 2.4 Structure of the disintegration continua

We conclude this section by studying the two disintegration continua $\mathcal{C}_{1\to2}(\mathbf{k}) = \{\epsilon_{\mathbf{k}-\mathbf{q}} + \hbar\omega_{\mathbf{q}}, \text{ for } \mathbf{q} \in \mathbb{R}^3\}$ and $\mathcal{C}_{1\to3}(\mathbf{k}) = \{\epsilon_{\mathbf{k}_1} + \epsilon_{\mathbf{k}_2} + \epsilon_{\mathbf{k}-\mathbf{k}_1-\mathbf{k}_2}, \text{ for } \mathbf{k}_1, \mathbf{k}_2 \in \mathbb{R}^3\}$ and their intersection as a function of $\mathbf{k}$. The lower edge of $\mathcal{C}_{1\to3}$ (red solid curve in Fig. 4) has a simple analytic expression [49]:

$$\epsilon_{\mathrm{th}}^{1\to3} = \begin{cases} 3\Delta, & \text{if } k \leq 3k_m^{(0)} \\ 3\epsilon_{k/3}, & \text{if } k > 3k_m^{(0)} \end{cases} . \tag{31}$$

This threshold is thus a three-body equivalent of the pair-breaking threshold (17). Its expression is obtained by a similar reasoning: as long as $k$ is below $3k_m^{(0)}$, the angular degrees of freedom can be used to accommodate three wave vectors $\mathbf{k}_1, \mathbf{k}_2, \mathbf{k}-\mathbf{k}_1-\mathbf{k}_2$ of norm $k_m^{(0)}$. Beyond this point the minimum of the continuum is reached by symmetry for equal wave vectors of norm $k/3$. The internal structure of the three-quasiparticle continuum, particularly complex due to the number of internal degrees of freedom, is studied in detail in Appendix A.

The lower edge of $\mathcal{C}_{1\to2}$ does not have in general an analytic expression because the eigenenergy $\omega_{\mathbf{q}}$ of the bosonic collective branch does not. For $k$ sufficiently close to the dispersion minimum $k_m^{(0)}$ (more precisely when the group velocity $\mathrm{d}\epsilon_k/\mathrm{d}k$ is below the speed of sound $c$ [51,61]), there exists a region $[k_{r1}, k_{r2}]$ where the continuum edge is reached in $q = 0$ such that $\epsilon_{\mathrm{th}}^{1\to2}(\mathbf{k}) = \epsilon_{\mathbf{k}}$. Outside this region the edge is reached at nonzero wave vector $q_{\mathrm{th}}$ (orange line in Fig. 4) and energy $\omega_{q_{\mathrm{th}}}$ (black line). For $k$ sufficiently close to the dispersion minimum $k_m^{(0)}$, this edge remains below $\epsilon_{\mathrm{th}}^{1\to3} = 3\Delta$ such that the resonant energies are divided in two sectors: sector A where only the $1 \to 2$ disintegration is energetically allowed and sector $B$ where both $1 \to 2$ and $1 \to 3$ disintegrations are allowed.

In the BCS regime (Fig. 4 is for $\mu/\Delta = 4$, $1/k_F a \simeq -0.91$) the bosonic branch has a termination point ($q_{\mathrm{sup}} \gtrsim 2k_F$ in the BCS regime) where it touches the pair-breaking continuum edge and disappears. This leads to a saturation of $q_{\mathrm{th}}$ at $k \approx k_{r2} + q_{\mathrm{sup}}$, and correspondingly to a rapid increase of $\epsilon_{\mathrm{th}}^{1\to2}$. This opens a new sector C where the $1 \to 3$ disintegration is resonant while the $1 \to 2$ is not. This sector however never contains the unperturbed quasiparticle energy $\epsilon_{\mathbf{k}}$.

We note that the quasiparticle spectral function $\rho_G(\mathbf{k}, \varepsilon) = \mathrm{Im}G(\mathbf{k}, \varepsilon + i0^+)$ is still not analytic within sector A, B, or C due to the internal structure of the disintegration continua, which contain angular points not shown on Fig. 4 (see for instance Appendix A). Still, the boundaries of those regions constitute major singularities of the spectral function, across which it cannot be fitted.

---

[2]All code used for our numerical calculations can be found on
https://github.com/hkurkjian/CodeArXiv_2111_04692

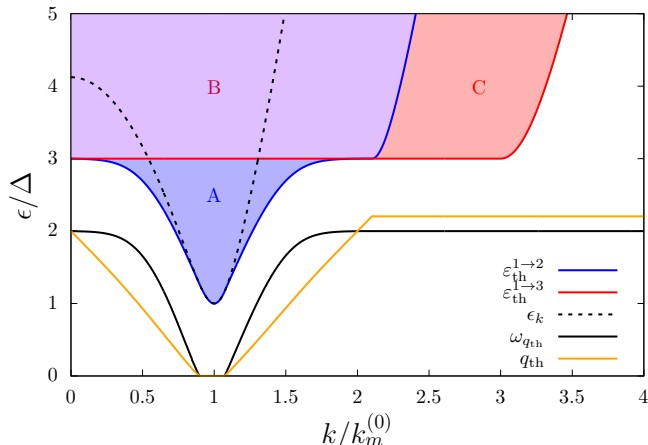

Figure 4: Diagram of the disintegration continua in function of the wave vector $k$ (in units of the mean-field dispersion minimum $k_m^{(0)} = \sqrt{2m\mu}/\hbar$) and energy $\epsilon$. Blue and purple area (regions A and B): the $1 \to 2$ disintegration continuum $\{\epsilon_{\mathbf{k}-\mathbf{q}} + \omega_{\mathbf{q}}$ for $\mathbf{q} \in \mathbb{R}^3\}$ caused by the poles of the pair propagator and bounded from below by $\epsilon_{\text{th}}^{1\to2}$ (blue solid curve). The orange and black solid lines are respectively the wave vector $q_{\text{th}}$ and energy $\omega_{q_{\text{th}}}$ of the boson emitted at the continuum threshold (which satisfy $\epsilon_{\mathbf{k}-\mathbf{q}_{\text{th}}} + \omega_{q_{\text{th}}} = \epsilon_{\text{th}}^{1\to2}$). Red and purple area (regions B and C): the $1 \to 3$ disintegration continuum $\{\epsilon_{\mathbf{k}_1} + \epsilon_{\mathbf{k}_2} + \epsilon_{\mathbf{k}-\mathbf{k}_1-\mathbf{k}_2}$ for $\mathbf{k}_1, \mathbf{k}_2 \in \mathbb{R}^3\}$ originating from the branch cut of the pair propagator and bounded from below by $\epsilon_{\text{th}}^{1\to3}$ (red solid curve). The two continua overlap in sector B (purple area), while sectors A and C are defined by $\epsilon_{\text{th}}^{1\to2}(k) < \epsilon < \epsilon_{\text{th}}^{1\to3}(k)$ and $\epsilon_{\text{th}}^{1\to3}(k) < \epsilon < \epsilon_{\text{th}}^{1\to2}(k)$ respectively. The dashed line shows the mean-field quasiparticle dispersion relation $\epsilon_{\mathbf{k}}$. Here, we took an interaction strength $\mu/\Delta = 4$ ($1/k_{\text{F}}a \simeq -0.91$).

## 3  Quasiparticle spectrum

### 3.1  Quasiparticle Green's function

Expressions (18) and (20) of the self-energies were all given in the particle-hole basis, where the single-particle propagator has both a positive and negative energy contribution, weighted by the Bogoliubov coefficients, as can be seen at the mean-field level in Eq. (4). The particle-propagator $G_{++}$ then consists of a mixture of quasiparticles with positive energy and quasiholes with negative energy. When studying the limit $g \to 0$ (Sec. 2.3) we explained that the Green's function takes a more intuitive form in the quasiparticle basis, after transformation with the mean-field Bogoliubov matrix $B_{\mathbf{k}}$. Here, we generalize this quasiparticle basis as the basis which diagonalizes the Green's function. This diagonalisation takes the form of a generalized Bogoliubov transformation:

$$\mathcal{B}(\mathbf{k},z)G^{-1}(\mathbf{k},z)\mathcal{B}^{\dagger}(\mathbf{k},z) = \begin{pmatrix} \mathcal{G}^{-1}(\mathbf{k},z) & 0 \\ 0 & -\mathcal{G}^{-1}(\mathbf{k},-z) \end{pmatrix}. \tag{32}$$

This defines the quasiparticle Green's function

$$\mathcal{G}^{-1}(\mathbf{k},z) = -z - \mathcal{D}(\mathbf{k},z) + \mathcal{E}(\mathbf{k},z), \tag{33}$$

and the energy functionals

$$\mathcal{D}(\mathbf{k}, z) = \frac{\Sigma_{++}(\mathbf{k}, z) + \Sigma_{--}(\mathbf{k}, z)}{2}, \tag{34}$$

$$\mathcal{E}(\mathbf{k}, z) = \sqrt{\left(\xi_{\mathbf{k}} + \frac{\Sigma_{--}(\mathbf{k}, z) - \Sigma_{++}(\mathbf{k}, z)}{2}\right)^2 + (\Delta - \Sigma_{+-}(\mathbf{k}, z))^2}. \tag{35}$$

The energy $\mathcal{E}(\mathbf{k}, z)$ plays the same role as the mean-field quasiparticle energy in Eq. (8), whereas $\mathcal{D}(\mathbf{k}, z)$ acts as a displacement of the reference energy (as long as its dependence on $z$ is omitted). The transfer matrix

$$\mathcal{B}(\mathbf{k}, z) = \begin{pmatrix} \mathcal{U}(\mathbf{k}, z) & \mathcal{V}(\mathbf{k}, z) \\ -\mathcal{V}(\mathbf{k}, z) & \mathcal{U}(\mathbf{k}, z) \end{pmatrix} \tag{36}$$

has the same structure as the mean-field transfer matrix, but contains altered Bogoliubov coefficients

$$\mathcal{U}(\mathbf{k}, z) = \sqrt{\frac{1}{2}\left(1 + \frac{\mathcal{X}(\mathbf{k}, z)}{\mathcal{E}(\mathbf{k}, z)}\right)}, \quad \mathcal{V}(\mathbf{k}, z) = \sqrt{\frac{1}{2}\left(1 - \frac{\mathcal{X}(\mathbf{k}.z)}{\mathcal{E}(\mathbf{k}, z)}\right)}, \tag{37}$$

with $\mathcal{X}(\mathbf{k}, z) = \xi_{\mathbf{k}} + (\Sigma_{--}(\mathbf{k}, z) - \Sigma_{++}(\mathbf{k}, z))/2$. Clearly, in the quasiparticle basis, it is enough to study the quasiparticle propagator $\mathcal{G}$, rather than the full matrix Green's function. We note that some usual experimental probes such as rf-spectroscopy [33] measure the Green's function in the particle-hole basis. However, key properties of the system, in particular dissipative properties, are directly sensitive to the quasiparticle spectrum.

## 3.2 Perturbative and self-consistent solutions

In terms of the quasiparticle Green's function, the equation on the quasiparticle energy takes the simple form

$$\mathcal{G}^{-1}(\mathbf{k}, z_{\mathbf{k}}) = 0. \tag{38}$$

Instead of solving this self-consistent equation on $z_{\mathbf{k}}$, it is tempting, given the form (33) of the quasiparticle Green's function, to perform a perturbative approximation on the energy functionals $\mathcal{D}$ and $\mathcal{E}$, that is, evaluate them in $z \to \epsilon_{\mathbf{k}} + i0^+$. Within this approximation, the solution of Eq. (38) reads

$$z_{\mathbf{k}} \simeq \mathcal{E}(\mathbf{k}, \epsilon_{\mathbf{k}} + i0^+) - \mathcal{D}(\mathbf{k}, \epsilon_{\mathbf{k}} + i0^+) \equiv E_{\mathbf{k}} - \frac{i\hbar}{2}\Gamma_{\mathbf{k}}, \tag{39}$$

where we have separated the real and imaginary part to distinguish between the corrected energy $E_{\mathbf{k}}$ and the damping rate $\Gamma_{\mathbf{k}}$ of the quasiparticles. In this perturbative approximation, the damping rate $\Gamma_{\mathbf{k}}$ is nonzero if the mean-field energy is inside one of the disintegration continua, such that at least one decay process is resonant. When the mean-field energy $\epsilon_{\mathbf{k}}$ is above $\epsilon_{\text{th}}^{1 \to 2}$ but below $\epsilon_{\text{th}}^{1 \to 3}$ (sector A), the only contribution to a finite lifetime comes from the boson-emission process $1 \to 2$ of Fig. 3a, while for $\epsilon_{\mathbf{k}} > \epsilon_{\text{th}}^{1 \to 3}$ (sector B), also the four-fermion process $1 \to 3$ of Fig. 3c lowers the lifetime. We note that $\epsilon_{\mathbf{k}}$ never enters sector C (the inequality $\epsilon_{\text{th}}^{1 \to 2} > \epsilon_{\mathbf{k}} > \epsilon_{\text{th}}^{1 \to 3}$ is never fulfilled), such that the fermionic disintegration process $1 \to 3$ never acts as the sole damping channel [49]. In a neutral gas, sector C thus has little practical importance, but the situation would be reversed in a charged superfluid where the bosonic branch acquires a large gap corresponding to the plasma frequency [19].

To look for the exact poles of the quasiparticle Green's function, Eq. (38) should be solved self-consistently (this is not to be confused with a self-consistent treatment of the self-energy $G^{(0)} \to G$ in Eqs. (11)–(13), which is beyond the scope of the present work). Close to the

minimum of the corrected fermionic branch, below the threshold energies $\epsilon_{\text{th}}^{1\to2}, \epsilon_{\text{th}}^{1\to3}$, a self-consistent solution of Eq. (38) can be found on the real axis, indicating well-defined quasi-particles with an infinite lifetime in this case. Once the solution enters the continuum, the self-consistent solution $z_{\mathbf{k}}$ obtains an imaginary part, and the quasiparticles are damped. Extracting this complex solution requires in principle an analytic continuation of $\mathcal{G}(z)$ through its branch cuts on the real axis (see Fig. 2). Unfortunately, in this problem, we do not have access to an analytic, or partially analytic [24, 52, 62], expression of the spectral density $\mathcal{G}(\mathbf{k}, \varepsilon + i0^+) - \mathcal{G}(\mathbf{k}, \varepsilon - i0^+)$ on which we could rely to extend the function to the lower-half complex plane. For this reason, we estimate the complex solution from its residuals just above the real axis in $\varepsilon + i0^+$. To do so, we fit the quasiparticle Green's function to a Lorentzian resonance

$$\mathcal{G}(\mathbf{k}, \varepsilon + i0^+) \underset{\text{fit in sector } X \text{ to}}{\longrightarrow} \frac{\Upsilon_{\mathbf{k}}^{(X)}}{z_{\mathbf{k}}^{(X)} - \varepsilon}, \tag{40}$$

and extract the fitted complex eigenenergy $z_{\mathbf{k}}^{(X)}$ and associated residue $\Upsilon_{\mathbf{k}}^{(X)}$. Since the Green's function shows a sharp singularity at the continuum thresholds $\epsilon_{\text{th}}^{1\to2}$ and $\epsilon_{\text{th}}^{1\to3}$, the fitting domain is always comprised in one of the analyticity sectors $X = A, B$ from Fig. 4. We note that the existence of several fitting sectors implies that two solutions $z_{\mathbf{k}}^{(A)}$ and $z_{\mathbf{k}}^{(B)}$ coexist for the same value of $k$. This reflects the structure of the analytic continuation, which has (at least) two separate Riemann sheets corresponding to the continuation through sector $A$ or $B$ [52, 62, 63] separated by branching points. This is thus not an artifact of our fitting strategy. We shall see in the discussion of Fig. 7 that the coexistence of the two solutions reflects a physical phenomenon, particularly near the point where Re$z_{\mathbf{k}}$ passes the fermionic threshold $\epsilon_{\text{th}}^{1\to3}$.

### 3.3 Numerical results

In Fig. 5 we show results of the quasiparticle spectrum at $\mu/\Delta = 4$ (thus in the shallow BCS regime, corresponding to $1/k_{\text{F}}a \simeq -0.9$). The imaginary part of the Green's function is exactly zero below the threshold energy $\epsilon_{\text{th}}^{1\to2}$. There, a real solution of $\mathcal{G}^{-1}(\mathbf{k}, z_{\mathbf{k}}) = 0$ can be found, represented as the part of the red curve below $\epsilon_{\text{th}}^{1\to2}$. It is clear that the minimum of the energy is shifted toward higher values of the wavenumber $k$ with respect to the mean-field energy, while the correction to the energy gap remains small. A more in-depth study of the low-energy spectrum can be found in the next section. Once the self-consistent solution reaches the first threshold $\epsilon_{\text{th}}^{1\to2}$ (on both sides of the minimum), the eigenenergy $z_{\mathbf{k}}$ becomes complex, which translates into a broadened peak in the spectral density. We keep track of the complex pole using the fitted energy $z_{\mathbf{k}}^{(A)}$ from Eq. (40). The real part perfectly continues the undamped solution (such that we represent it by the same red curve on the top panel), illustrating the efficiency of our fitting strategy. Further away from the dispersion minimum, the resonance reaches the second threshold $\epsilon_{\text{th}}^{1\to3}$ and the red curve switches from $z_{\mathbf{k}}^{(A)}$ to $z_{\mathbf{k}}^{(B)}$ to describe the resonance in region $B$. The transition is again very smooth, and only translates into a small kink in the damping rate (bottom panel). This is in stark contrast to previous results of Ref. [48] taking into account only the $1 \to 2$ disintegration, where the damping rate is sharply peaked when the eigenenergy reaches the threshold $\epsilon_{\text{th}}^{1\to3}$. This peaked behavior is washed out by including the now resonant $1 \to 3$ process in the self-energy, which sharply reduces the quasiparticle lifetime. At larger $k$, the damping rate peaks around $\hbar k \simeq 5.5\sqrt{2m\Delta}$, after which it exhibits a large $1/k$ tail. In the limit $k \to \infty$, where the interaction between the quasiparticle and the rest of the superfluid becomes negligible, the quasiparticle energy tends to the kinetic energy of a free fermion: $z_{\mathbf{k}} \sim \hbar^2 k^2/2m$.

The self-consistent eigenenergy can be compared to the perturbative result of Eq. (39),

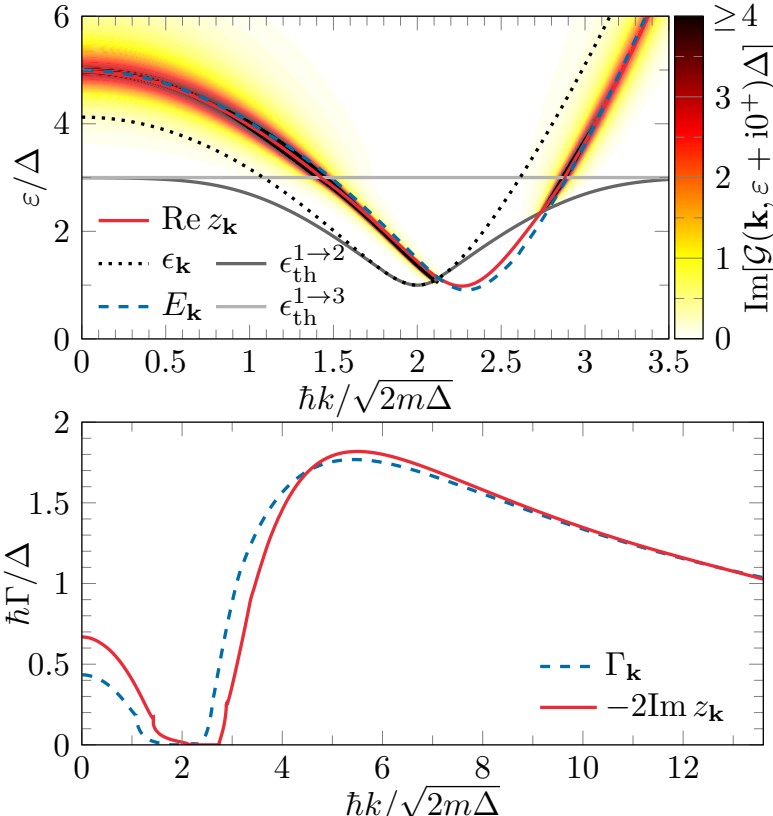

Figure 5: Quasiparticle spectrum in the BCS regime ($\mu/\Delta = 4$, $1/k_{\mathrm{F}}a \simeq -0.9$). The quasiparticle spectral density $\mathrm{Im}[\mathcal{G}(\mathbf{k}, \varepsilon + i0^+)]$ is shown as a function of the wavenumber $k$ and energy $\varepsilon$ in units of the order parameter $\Delta$. This is compared to the self-consistent energy (red line found by solving Eq. (38) in the undamped region, and by fitting the Green's function to Eq. (40) elsewhere), the perturbative energy given by Eq. (39) (blue dashed line), and the mean-field energy (black dotted line). The threshold energies $\epsilon_{\mathrm{th}}^{1\to2}$ (dark grey) and $\epsilon_{\mathrm{th}}^{1\to3}$ (light grey) are also shown to distinguish the different regions of possible resonances. In the lower panel, the imaginary part of the eigenenergy is plotted, showing the damping rate of the quasiparticles.

which is shown as a blue dashed line in Fig. 5. Both results remain close to each other, although the energy gap is lowered in the perturbative case with respect to the self-consistent solution. Most notably, the perturbative spectrum incorrectly predicts a finite lifetime of the quasiparticles at their energy minimum. This is due to the fact that the resonance condition in this case is controlled by the mean-field energy $\epsilon_{\mathbf{k}}$, which does not account for the shift in the energy minimum. This is resolved with the self-consistent solution, which accurately predicts well-defined quasiparticles at the energy minimum.

In order to reveal the underlying mixing of the particle and hole channels contributing to the quasiparticle spectrum, we show in Fig. 6 the altered Bogoliubov coefficients of Eq. (37). Similar to the BCS case, before the dispersion minimum $\mathcal{V}$ is close to unity and $\mathcal{U}$ is small, indicating that the quasiparticles resemble hole excitations, while at higher $\mathbf{k}$ values the situation is reversed, such that the quasiparticles are well approximated by particle excitations. The region where the mixing is largest corresponds to the dispersion minimum and is consequently shifted to higher $k$ values compared to the BCS prediction. Note that when $z_{\mathbf{k}}$ is complex the self-consistent values of $\mathcal{U}$ and $\mathcal{V}$ are evaluated in $\mathrm{Re}\, z_{\mathbf{k}}$, rather than in $z_{\mathbf{k}}$ itself, as we don't

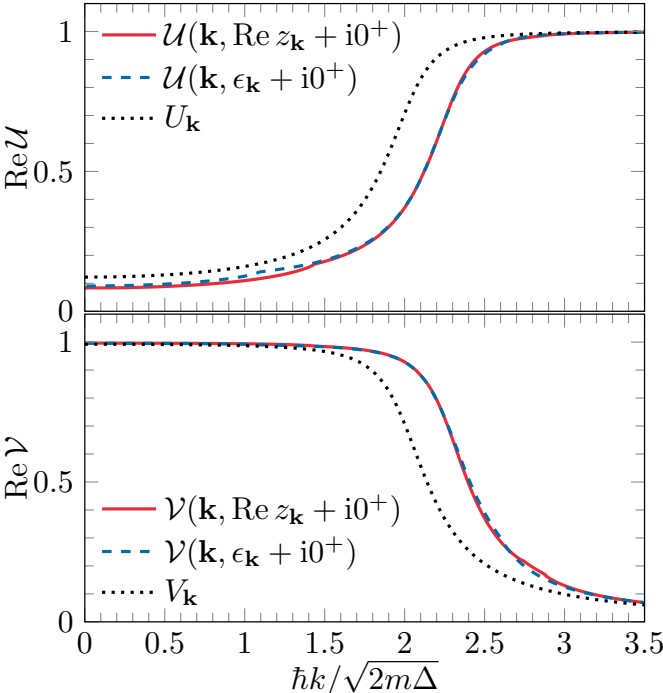

Figure 6: Bogoliubov coefficients in the BCS regime ($\mu/\Delta = 4$, $1/k_\mathrm{F}a \simeq -0.9$). The altered Bogoliubov weights defined in Eq. (37) are shown in function of the wavenumber using the eigenenergy from different approximations used in our study, namely the self-consistent (red line) and perturbative (blue dashed line) solutions. The corrected coefficients are shifted with respect to the BCS result (black dotted line), in the same way as the minimum of the dispersion (see Fig. 5).

have access to the analytical continuation of the self-energy. This approximation works well as long as the imaginary part of $z_\mathbf{k}$ remains small.

We perform a similar analysis at unitarity in Fig. 7, where the quasiparticle spectral density is shown and compared with the eigenenergies from the self-consistent and perturbative methods. Here, however, we notice a more abrupt transition when the resonance reaches the second threshold $\epsilon_\mathrm{th}^{1\to3}$. To make this clear, we show on Fig. 8 a cross section of the Green's function at $\hbar k = 2.16\sqrt{2m\Delta}$ where we noticed a marked angular point in $\epsilon = \epsilon_\mathrm{th}^{1\to3}$. At the same time, the resonance in both sectors $A$ and $B$ loses its Lorentzian behavior (in other words, the fit in Eq. (40) loses accuracy), as visible in the large shoulders on the side of the resonance peaks in Fig. 8. Incidentally, the perturbative method predicts a large damping rate in this regime (blue dashed curve on the bottom panel of Fig. 7). Overall, this behavior indicates that the quasiparticle is no longer a well defined object when its energy approaches $3\Delta$ at strong-coupling. This is in qualitative agreement with the results of Ref. [48].

One possibility to account for the increasing complexity of the quasiparticle Green's function, and in particular to model the shoulders that the resonance develops in this regime, is to use a fit function describing two interfering resonances:

$$\mathcal{G}(\mathbf{k}, \varepsilon + i0^+) \underset{\text{fit in sector } X \text{ to}}{\longrightarrow} \frac{\Upsilon_\mathbf{k}^{(X,1)}}{z_\mathbf{k}^{(X,1)} - \varepsilon} + \frac{\Upsilon_\mathbf{k}^{(X,2)}}{z_\mathbf{k}^{(X,2)} - \varepsilon}, \tag{41}$$

where $X = A, B$ and the second poles (not explicitly shown in Fig. 7) have a lower spectral weight than the first ones $|\Upsilon_\mathbf{k}^{(X,2)}| < |\Upsilon_\mathbf{k}^{(X,1)}|$ by convention. This second pole in the energy sector $A$ ($\epsilon_\mathrm{th}^{1\to2} < \varepsilon < \epsilon_\mathrm{th}^{1\to3}$) can be seen as a remnant of the first pole from sector $B$ ($\epsilon_\mathrm{th}^{1\to3} < \varepsilon$)

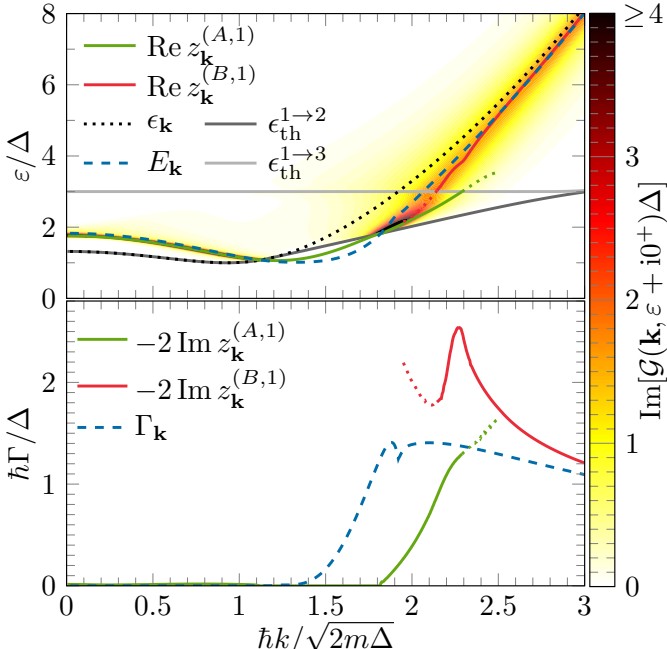

Figure 7: Quasiparticle spectrum at unitarity ($\mu/\Delta \simeq 0.86$). To correctly account for different possible branches, the energy domain for which we fit the quasiparticle Green's function is split up for each value of the wavenumber $k$. Three different regions can be recognized, separated by the threshold energies $\epsilon_{\text{th}}^{1\to2}$ (dark grey) and $\epsilon_{\text{th}}^{1\to3}$ (light gray). In the first region ($\varepsilon < \epsilon_{\text{th}}^{1\to2}$) the quasiparticle propagator is real, and a real self-consistent pole $z_{\mathbf{k}}$ can be found (green line). Above the threshold $\epsilon_{\text{th}}^{1\to2}$, the complex eigenenergy is computed from a fit as in Eq. (41), using only energy values $\epsilon_{\text{th}}^{1\to2} < \varepsilon < \epsilon_{\text{th}}^{1\to3}$ (green line, sector A), or $\epsilon_{\text{th}}^{1\to3} < \varepsilon$ (red line, sector B). We furthermore compare the self-consistent solutions with the perturbative (blue dashed line) and mean-field energy (black dotted line). In the lower panel, we show the imaginary part of the eigenenergy, with the same color code.

and vice-versa. Note that the main fitted solutions $z_{\mathbf{k}}^{(A,1)}$ (in green) and $z_{\mathbf{k}}^{(B,1)}$ (in red) continue to formally exist outside their respective energy sector, with however a decreasing fitting accuracy, and less physical significance [52, 62]. For this reason, we display the eigenfrequencies as dotted lines outside their respective energy sector on Fig. 7. We note an important mismatch between $z_{\mathbf{k}}^{(A,1)}$ and $z_{\mathbf{k}}^{(B,1)}$, in the real and especially the imaginary part, which reflects the repulsion exerted on the resonance peak by the $3\Delta$ threshold.

This non trivial behavior near $\epsilon_{\text{th}}^{1\to3}$ is missed by the perturbative solution which incorrectly predicts a smooth entry into the fermionic $1 \to 3$ continuum (although with a degraded quality factor $E_{\mathbf{k}}/\Gamma_{\mathbf{k}} \approx 1$). Another discrepancy between the perturbative and self-consistent energies is visible at low $k$, in the decreasing part of the dispersion: there, the mean-field energy is at the disintegration threshold $\epsilon_{\mathbf{k}} = \epsilon_{\text{th}}^{1\to2}$, such that perturbative approach predicts undamped quasiparticles. Conversely, the self-consistent solution $z_{\mathbf{k}}^{(A)}$ is pushed up in energy and thus acquires a (small) nonzero damping rate.

## 3.4 Low-energy properties

The previous section has shown that the quasiparticles corresponding to the exact zeros of Eq. (38) remain well-defined around their energy minimum everywhere in the BCS-BEC crossover. It is thus useful to explicitly examine the low-energy properties of the quasiparticle

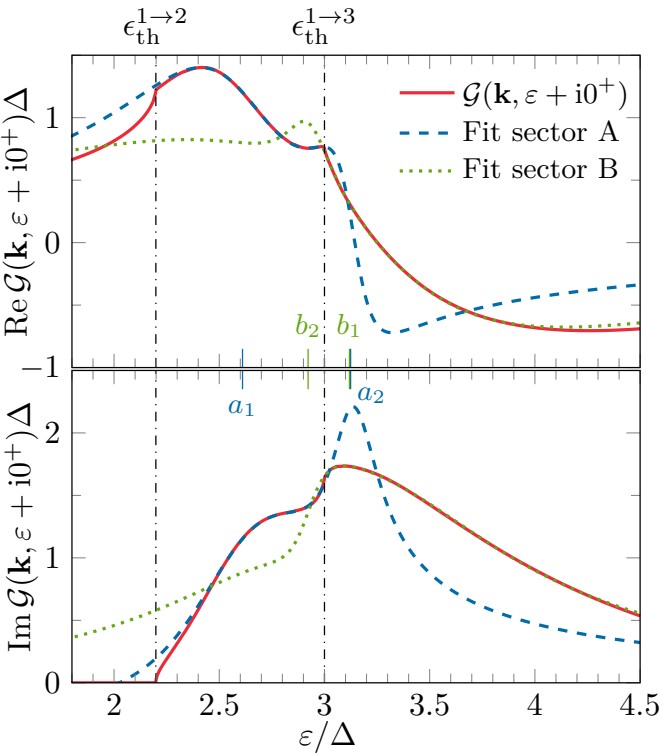

Figure 8: Cross section of the real (top panel) and imaginary (bottom panel) part of the quasiparticle Green's function at unitarity and $\hbar k = 2.16\sqrt{2m\Delta}$. In order to provide a good fit of the propagator, we split up the energy domain in resonance domains delimited by the threshold energies $\epsilon_{\text{th}}^{1\to 2}$ and $\epsilon_{\text{th}}^{1\to 3}$ shown as black dash-dotted lines. The fitted function according to Eq. (41) is shown as a blue dashed line below $\epsilon_{\text{th}}^{1\to 3}$ (sector A) and as a green dotted line above (sector B). The real parts of the eigenenergies found from the fit are indicated on the axis with $a_j \equiv \text{Re}\, z_{\mathbf{k}}^{(A,j)}$ and $b_j \equiv \text{Re}\, z_{\mathbf{k}}^{(B,j)}$

branch, which we do in this last section.

In order to extract experimentally relevant properties from the quasiparticle branch, we fit a quadratic dispersion around the minimum of the eigenenergy

$$\epsilon_k^{\text{fit}} = \epsilon^* + \frac{\hbar^2 (k - k_{\text{m}}^*)^2}{2m^*}, \tag{42}$$

and study the effective energy gap $\epsilon^*$, location of the minimum $k_{\text{m}}^*$, and effective mass $m^*$ for different values of the interaction. The results are shown in Figs. 9 and 10.

In Fig. 9 we show the (squared) shift of the dispersion minimum with respect to the mean-field value $k_{\text{m}}^{(0)}$. This shift is strictly positive throughout the BEC-BCS crossover (as long as $k_{\text{m}}^* \neq 0$), such that the dispersion minimum is reached at shorter wavelengths. In the BCS limit, our numerical results recover the Hartree shift $(k_{\text{m}}^*)^2 - (k_{\text{m}}^{(0)})^2 = -4k_{\text{F}}^3 a/3\pi$, as shown by the dotted asymptote. Our inconsistent $t$-matrix self-energy however does not capture the next-to-leading order correction to the Hartree shift obtained by Galitskii [59, 66] (dashed line in Fig. 9). The appearance of the Hartree shift can be understood heuristically by retaining only the leading order term in the expansion Eq. (23) of the pair propagator for $g \to 0$: $\Gamma_{ss'} = -g\delta_{ss'} + \mathcal{O}(g^2)$. Then, the off-diagonal elements of the self-energy become zero, while $\Sigma_{++}(k) = g/\beta V \sum_{\mathbf{k},n} G_{++}^{(0)}(k)$. This sum can be performed analytically, to find

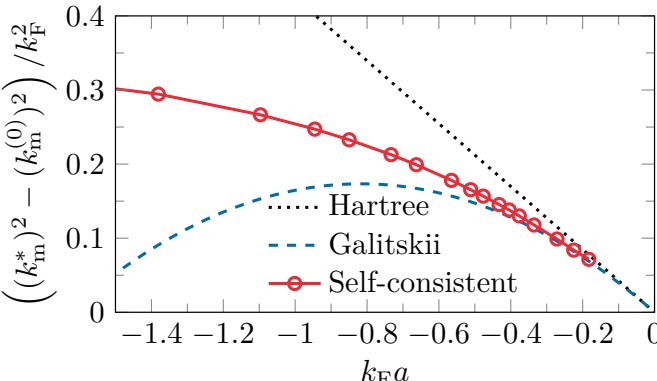

Figure 9: The quadratic shift of the energy minimum compared to the mean-field value. Our self-consistent calculation of the shift (red circles) is compared to the Hartree [56, 57, 64, 65] (black dotted line) and Galitskii [59, 66] (blue dashed line) results in the BCS limit.

$\Sigma_{++} = -gn/2$, with $n$ the total fermion density. This is translated in the fermionic energy by a shift of the chemical potential $\mu \to \mu - gn/2 = \mu - 4\epsilon_F k_F a/3\pi$, better known as the Hartree shift [56, 57, 64, 65]. Note that although this Hartree shift tends to 0 in the BCS limit, it becomes much larger than the gap (which vanishes exponentially with $k_F a$). Near the dispersion minimum, this forbids a linearisation of Eq. (38) where $\Sigma$ would be treated as an infinitesimal [48]. Although in general it is not possible to separate the contributions of the different disintegration processes, the appearance of the Hartree shift in the BCS limit can be understood to be mainly coming from $\Sigma^{bc}$, as it is not present when only including the $1 \to 2$ process [48].

Looking then at the energy gap in Fig. 10, we observe a fairly small correction throughout the BCS-BEC crossover. We underline that this figure should be interpreted by keeping in mind the difference between the energy gap $\epsilon^*$ and the order parameter $\Delta$. While most theoretical works (see e.g. Refs. [55–57]) focus on the order parameter, which is a variational parameter determining the superfluid phase transition, experimental results usually measure (twice) the energy gap (see e.g. Refs. [36, 41]), the energy needed to break a condensed pair. While both are the same on the mean-field level, this is no longer obviously true when fluctuations are taken into account.

In the BCS limit, the corrected gap tends to the mean-field gap, $\epsilon^*/\Delta \to 1$. This appears as a serious limitation of our inconsistent $t$-matrix self-energy, since Gor'kov and Melik-Barkhudarov [55] have predicted a lowering of the order parameter of a factor $\approx 2.2$, while experimental results have reported a correction of the same magnitude on the energy gap [38]. To account for this reduction of the gap, it is necessary to add diagrams to the quasiparticle self-energy to include a similar correction as in Ref. [55]. For a consistent theory, the equation for the order parameter should then also be modified. In this case, it is still an open question if the corrected gap and order parameter are equal, even in the weak-coupling limit. In the BCS limit, we also note that the perturbative result for $\epsilon^*$ differs much from the self-consistent one. This can be seen as a consequence of the Hartree-shift: the energy displacement $\text{Re}(z_k) - \epsilon_k$ near the dispersion minimum is large compared to $\Delta$ (as visible on Fig. 5), such that the replacement $z_k \to \epsilon_k + i0^+$ in Eq. (39) has a large effect on the energy correction. At unitarity, we find the gap to be larger than the mean-field prediction $\epsilon^* \simeq 1.06\Delta$. This contrasts with our previous study [48] limited to the bosonic decay $1 \to 2$, and with experimental results observing a reduction of the gap [38, 41]. Possible explanations of this quantitative mismatch are the non-self-consistent definition of the self-energy and the omission of the Gorkov-Melik

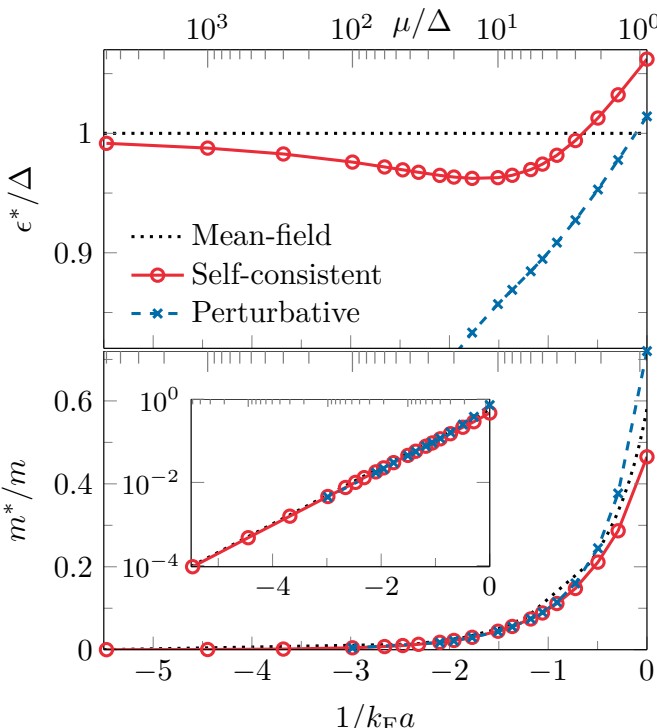

Figure 10: The quasiparticle gap (top panel) and effective mass (bottom panel) in function of the interaction $1/k_\mathrm{F}a$ (bottom axis) or $\mu/\Delta$ (top axis). The red circles show the self-consistent solutions, which are compared to the perturbative results (blue crosses) and the mean-field values (black dotted line). It is clear that the perturbative calculation leads to wrong results in the BCS limit, as it is not possible to treat the self-energy $\Sigma$ as a small correction.

Barkhudarov contribution to the self-energy.

Finally, it is useful to examine the low-energy spectrum in detail in the BCS limit, which can be seen in Fig. 11 for $\mu/\Delta = 100$ ($1/k_\mathrm{F}a \simeq -3$). There, as explained above, the self-energy mainly produces a shift of the energy minimum, while the correction to the gap remains small. We can align the computed eigenenergies by shifting the wavenumber with respect to the location of the minimum $k_\mathrm{m}^*$, to reveal that the structure of $z_\mathbf{k}$ remains close to the BCS energy (this can also be seen in Fig. 10, as both the gap and effective mass tend to the mean-field result in the BCS limit). Due to the large shift, the self-consistent energy enters the continuum at $\mathbf{k}$-values where the threshold $\epsilon_\mathrm{th}^{1\to2}$ is very close to $\epsilon_\mathrm{th}^{1\to3}$, such that the energy interval in sector A is small (see Fig. 4). Therefore, the damping rate rapidly increases as both the $1 \to 2$ and $1 \to 3$ processes are resonant. The perturbative approach, however, breaks down in the BCS limit as it fails to capture the Hartree shift, as explained above. Most notably, the perturbative approximation predicts a finite lifetime of the quasiparticle excitations at the energy minimum, as the resonance condition is controlled by the mean-field energy $\epsilon_\mathbf{k}$. For this reason, we have to shift the perturbative damping rate by $k_\mathrm{m}^{(0)}$ instead of $k_\mathrm{m}^*$ used for the eigenfrequency to be able to compare with the self-consistent solution in Fig. 11. Doing this, we see that the damping rate remains small when the BCS energy enters sector A and starts to grow rapidly when it reaches the second threshold energy $\epsilon_\mathbf{k} > \epsilon_\mathrm{th}^{1\to3}$ (sector B).

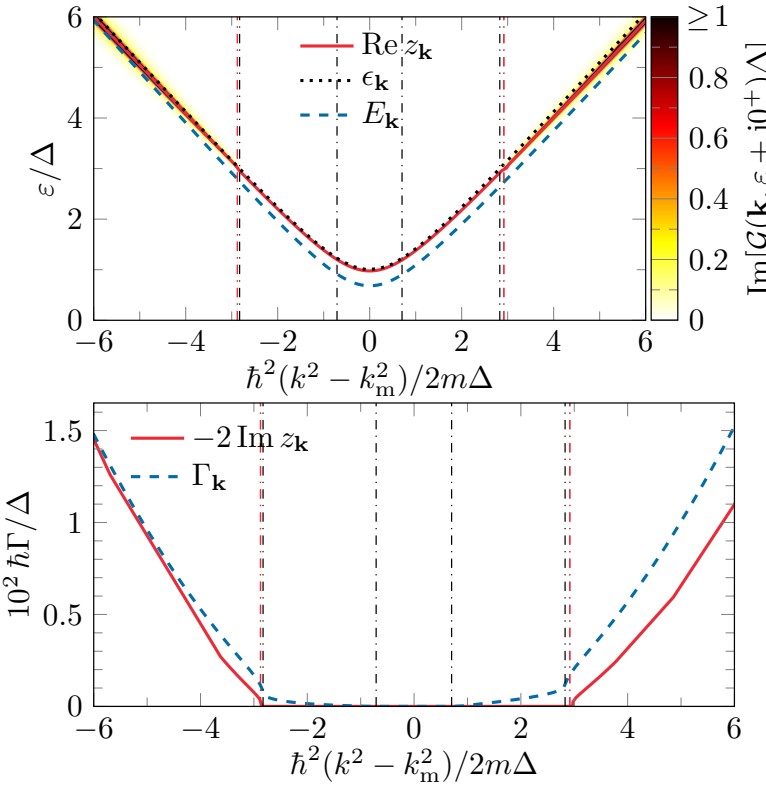

Figure 11: Quasiparticle spectrum in the BCS regime ($\mu/\Delta = 100$, $1/k_F a \simeq -3$). The eigenenergies and spectral density are shifted to facilitate the comparison between the different curves. The self-consistent solution is shown in red, and both its real (top panel) and imaginary part (bottom panel) are shifted with respect to $k_m^*$ computed from $z_\mathbf{k}$. The perturbative result (blue dashed line) incorrectly predicts a finite quasiparticle lifetime at the dispersion minimum, but to be able to compare with the self-consistent result, the real part $E_\mathbf{k}$ is shifted with respect to $k_m^*$, while the damping rate $\Gamma_\mathbf{k}$ is shifted to $k_m^{(0)}$. The black dotted line is the mean-field energy, and the vertical dash-dotted lines depict the shifted $k$-values where the mean-field energy $\epsilon_\mathbf{k}$ and self-consistent energy $z_\mathbf{k}$ cross the threshold energies respectively in black and red.

## 4 Conclusion

We have performed an in-depth study of the quasiparticle spectrum in a superfluid Fermi gas and identified the disintegration processes described by the $t$-matrix self-energy. We find evidence of multiple excitation branches at strong coupling close to the threshold energy where the four-fermion processes of disintegration into 3 quasiparticles become resonant. Using momentum-resolved rf-spectroscopy [42], the quasiparticle spectrum can directly be probed, such that our results could be experimentally verified. At weak-coupling, our approach captures the Hartree shift of the dispersion minimum, but not the Gor'kov-Melik Barkhudarov correction to the gap. Generalizing the correction of Ref. [55] to the quasiparticle spectrum thus appears as a necessary continuation of the present study.

Although the calculation was done within the framework of ultracold fermionic gases, similar decay processes occur in other quantum many-body systems where both phononic and rotonic excitations are present [5, 6, 10–15]. In superconductors, besides the intrinsic processes considered here, the quasiparticle lifetime is limited also by extraneous processes

such as impurities scattering [26] and emission of lattice phonons [10]. However, the $3\Delta$ threshold should still be visible, and the $1 \to 3$ disintegration process should be the main intrinsic disintegration process of the electron gas, owing to the gapped nature of the collective plasma branch.

## Acknowledgments

Discussions with C. A. R. Sá de Melo are gratefully acknowledged. SVL was supported by a Fellowship of the Belgian American Educational Foundation. The computational resources and services used in this work were provided by the HPC core facility CalcUA of the Universiteit Antwerpen, and VSC (Flemish Supercomputer Center), funded by the Research Foundation - Flanders (FWO) and the Flemish Government.

## A    Integration over the internal structure of the $1 \to 3$ disintegration continuum

In this appendix we study the internal structure of the fermionic disintegration process $1 \to 3$ (Fig. 3c), in relation to the numerical evaluation of $\Sigma^{\text{bc}}$ in Eq. (20). Rather than using the symmetric formulation as in Eq. (22), the form of the integral incites us to pair up two of the emitted quasiparticles. The process is then resonant for a given value of the fermion momentum $k$ and energy $z_k$ if one can find $\mathbf{q}$ and $\omega_q$ such that:

$$z_k - \omega_q = \epsilon_{\mathbf{k}-\mathbf{q}}, \tag{43}$$

where $\omega_q$ has to be inside the pair-breaking continuum $[\epsilon_c(q), +\infty[$. (Everywhere in this appendix, energies and momenta are in units of $\Delta$ and $k_\Delta = \sqrt{2m\Delta}/\hbar$, and $k_0 = \sqrt{2m\mu}/\hbar$ denotes the mean-field dispersion minimum). Whether this resonance condition is met or not has a huge impact on the integrand of Eq. (20), in particular on its imaginary part. This motivates a precise identification of the resonance domains.

Numerically, we integrate successively over the angle $u = \mathbf{k} \cdot \mathbf{q}/kq$ (the integral is done analytically as explained in Appendix B), then over $\omega_q$ and finally over $q$. As we integrate over $u$, the energy $\epsilon_{\mathbf{k}-\mathbf{q}}$ of the unpaired emitted quasiparticle goes from $\epsilon_{k-q}$ to $\epsilon_{k+q}$ (black solid curves in Fig. 12), either monotonically (for $q < |k - k_0|$ or $q > k + k_0$) or non-monotonically (for $|k - k_0| < q < k + k_0$) if it passes through the energy minimum (horizontal dotted line in Fig. 12). Depending on the value of $z_k - \omega_q$, there can then be either 0, 1 or 2 resonance angles. We then integrate over $\omega_q$, which corresponds to a descending vertical path on Fig. 12, as shown by the red arrow. Along the integration, the number of resonance angles changes if $\omega_q$ passes through $z_k - \epsilon_{k-q}$ or $z_k - \epsilon_{k+q}$ and if $\omega_q$ passes $z_k - 1$ while in the blue (2R) region. To reach a good precision on the integral, we split it at those values. Depending on the values of $k, z_k$ and the last integration variable $q$, there are five possible configurations in which the integral over $\omega_q$ may be split; they are denoted by greek letters ($\alpha, \beta, \gamma, \delta$ and $\epsilon$), described in Tab. 1, and shown in colors on the right panel of Fig. 12 in function of $q$ and $z_k - 2$. In the example shown by the red arrow of Fig. 12, there are 0 resonance angles from $\omega_q = \epsilon_c(q)$ to $z_k - \epsilon_{k+q}$, 1 from $z_k - \epsilon_{k+q}$ to $z_k - \epsilon_{k-q}$, 2 from $z_k - \epsilon_{k-q}$ to $z_k - 1$, and again 0 from $z_k - 1$ to $+\infty$, which corresponds to the configuration $\epsilon$ in Tab. 1.

The succession of the $\alpha, \beta, \gamma, \delta$ or $\epsilon$ configurations as we finally integrate from $q = 0$ to $+\infty$ gives rise to one of the superconfigurations denoted by capital letters from A to J and described in Tabs. 2 and 3 (for $k < k_0$ and $k > k_0$ respectively). The boundary values of $q$ where

the (greek-letter) configuration changes are then given in Tabs. 4 and 5 (again for $k < k_0$ and $k > k_0$ respectively). Note that in all the zoology of superconfigurations, the set of resonant wave numbers $q$ is always either the empty set or a (connected) interval (which can easily be read on Tabs. 2 and 3 as the interval between the lower bound of the first configuration and the upper bound of the last one, excluding the 0 configuration). Numerically, the splitting of the integral over $q$ at those boundaries is thus less crucial than for the integral over $\omega_q$, except at the boundaries where the resonance totally disappears.

Finally the remarkable values of $k$ at which the energy lines cross or anticross (the vertical lines on Fig. 13) are gathered in Tab. 7. To each interval between successive $k_i$ values corresponds a type of diagram (denoted by Roman numbers) similar to Fig. 12.

Fig. 13 shows the superconfigurations in colors in function of $k$ and $z_k$. The energy lines which separate them in the $k, z_k$ plane (among which the lower edge of the continuum $\epsilon_{\text{th}}^{1\to3}$) are given in Tab. 6. The only energy line posing a slight difficulty is $\omega_2'$ (which separates superconfigurations J and G, see Fig. 13). At this energy, the displaced pair-breaking continuum is tangent to the function $q \mapsto \epsilon_{k-q}$, such that

$$\omega_2' - 2\epsilon_{q_{\text{tg}}/2} = \epsilon_{k-q_{\text{tg}}}, \tag{44}$$

$$v_{q_{\text{tg}}/2} = v_{k-q_{\text{tg}}}, \qquad \text{with } v_k = \partial\epsilon_k/\partial k, \tag{45}$$

$$k_4 \leq k \leq k_3, \text{ type IV } (k_0 = 2, k = 0.92)$$

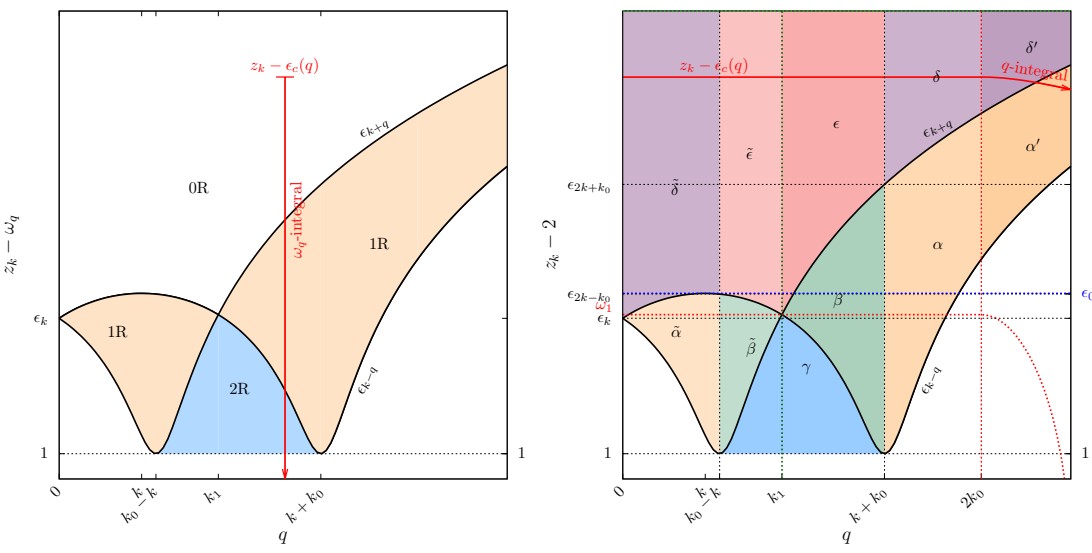

Figure 12: Resonance diagrams and integration over $\omega_q$ and $q$. The left panel shows the number of resonance angles in function of $q$ and $\omega_q$ (the $y$-axis is $z_k - \omega_q$): either 0 (white region), 1 (yellow region) or 2 (blue region) resonance angles, depending on whether the function $u \mapsto \epsilon_{\mathbf{k-q}}$ is monotonous or not, and whether $z_k - \omega_q$ lies within its energy-range or not (black solid line: value $\epsilon_{k\mp q}$ of the function in $u = \pm1$). As we then integrate over $\omega_q$ in the pair-breaking continuum ($z_k - \omega_q$ varies from $z_k - \epsilon_c(q)$ to $-\infty$, see the red vertical arrow on the left panel), the number of resonance angles changes according to a configuration $\alpha$, $\beta$, $\gamma$, $\delta$ or $\epsilon$ (see Tab. 1) shown in color on the right panel. As we finally integrate over $q$ (red solid curve on the right panel), the succession of greek-letter configurations gives rise to one of the superconfiguration of Tabs. 2 and 3. This superconfiguration itself changes when $z_k$ passes one of the energy lines shown as thin dashed horizontal curves. This figure is drawn for $k_4 < k < k_3$ (vertical sector number IV in Fig. 13) and $k_0 = 2$.

Table 1: Resonance configurations of the integral over $\omega_q$ (at fixed $k$, $z_k$ and $q$ but after integration over $u$). The table reads as follows: the configuration $\beta$ has 1 resonance angle (1R) for $2 < \omega_q < z_k - \epsilon_{k-q}$, 2 angles (2R) for $z_k - \epsilon_{k-q} < \omega_q < z_k - 1$ and 0 angle (0R) for $\omega_q > z_k - 1$. The configurations symbolized by the prime letters $\alpha', \ldots \epsilon'$ are deduced from $\alpha, \ldots \epsilon$ by changing the value of the continuum threshold: $2 \rightarrow \epsilon_c(q > 2k_0)$. The tilde configuration $\tilde{\alpha}$, $\tilde{\beta}$, $\tilde{\delta}$ and $\tilde{\epsilon}$ are obtained by swapping $\epsilon_{k-q}$ and $\epsilon_{k+q}$ in the energy boundaries of $\alpha$, $\beta$, $\delta$, $\epsilon$.

| Configurations of the $\omega_q$ integral | | | | | | | |
|---|---|---|---|---|---|---|---|
| Name | Energy boundaries | | | | 0,1 or 2 resonances | | |
| 0 | 2 | | | | 0R | | |
| $\alpha$ | 2 | $z_k - \epsilon_{k-q}$ | | | 1R | 0R | |
| $\beta$ | 2 | $z_k - \epsilon_{k-q}$ | $z_k - 1$ | | 1R | 2R | 0R |
| $\gamma$ | 2 | $z_k - 1$ | | | 2R | 0R | |
| $\delta$ | 2 | $z_k - \epsilon_{k+q}$ | $z_k - \epsilon_{k-q}$ | | 0R | 1R | 0R |
| $\epsilon$ | 2 | $z_k - \epsilon_{k+q}$ | $z_k - \epsilon_{k-q}$ | $z_k - 1$ | 0R 1R 2R 0R | | |

Table 2: For $k < k_0$, table of the resonance superconfigurations for the integration over $q$ (at fixed $k$ and $z_k$ and after integration over $u$ and $\omega_q$). The first column is the name of the configuration in Roman letters, the second column is the list of $p + 1$ boundaries values of $q$ (see Table 4 for their meaning) splitting the integration interval $[0, +\infty[$ into $p$ domains, and the third column is the configuration itself: a list of $p$ greek letters (plus the final configuration 0 not written explicitly), each one denoting the shape of the resonance in the subintegral over $\omega_q$ (see Table 1).

| Superconfigurations of the $q$-integral for $k < k_0$ | | |
|---|---|---|
| Name | Boundaries | Configuration |
| A | $0 \quad q_1 \quad k_- \quad q_2 \quad\quad q_2' \quad k_+ \quad q_3'$ | $0 \quad \tilde{\alpha} \quad\quad \tilde{\beta} \quad \gamma \quad \beta \quad \alpha \text{ or } \alpha'$ |
| B | $0 \quad q_1' \quad k_- \quad q_2 \quad\quad q_2' \quad k_+ \quad q_3'$ | $\tilde{\delta} \quad \tilde{\alpha} \quad\quad \tilde{\beta} \quad \gamma \quad \beta \quad \alpha \text{ or } \alpha'$ |
| B′ | $0 \quad q_1 \quad k_- \quad q_2' \quad q_c \quad q_2 \quad k_+ \quad q_3'$ | $0 \quad \tilde{\alpha} \quad \tilde{\beta} \quad \tilde{\epsilon} \quad \epsilon \quad \beta \quad \alpha \text{ or } \alpha'$ |
| C | $0 \quad k_- \quad q_1' \quad q_2 \quad\quad q_2' \quad k_+ \quad q_3'$ | $\tilde{\delta} \quad \tilde{\epsilon} \quad\quad \tilde{\beta} \quad \gamma \quad \beta \quad \alpha \text{ or } \alpha'$ |
| C′ | $0 \quad q_1' \quad k_- \quad q_2' \quad q_c \quad q_2 \quad k_+ \quad q_3'$ | $\tilde{\delta} \quad \tilde{\alpha} \quad \tilde{\beta} \quad \tilde{\epsilon} \quad \epsilon \quad \beta \quad \alpha \text{ or } \alpha'$ |
| C″ | $0 \quad q_1 \quad q_2' \quad k_- \quad q_c \quad q_2 \quad k_+ \quad q_3'$ | $0 \quad \tilde{\alpha} \quad \tilde{\delta} \quad \tilde{\epsilon} \quad \epsilon \quad \beta \quad \alpha \text{ or } \alpha'$ |
| D | $0 \quad k_- \quad q_c \quad q_2 \quad q_1' \quad q_2' \quad k_+ \quad q_3'$ | $\tilde{\delta} \quad \tilde{\epsilon} \quad \epsilon \quad \beta \quad \gamma \quad \beta \quad \alpha \text{ or } \alpha'$ |
| D′ | $0 \quad k_- \quad q_1' \quad q_2' \quad q_c \quad q_2 \quad k_+ \quad q_3'$ | $\tilde{\delta} \quad \tilde{\epsilon} \quad \tilde{\beta} \quad \tilde{\epsilon} \quad \epsilon \quad \beta \quad \alpha \text{ or } \alpha'$ |
| D″ | $0 \quad q_1' \quad q_2' \quad k_- \quad q_c \quad q_2 \quad k_+ \quad q_3'$ | $\tilde{\delta} \quad \tilde{\alpha} \quad \tilde{\delta} \quad \tilde{\epsilon} \quad \epsilon \quad \beta \quad \alpha \text{ or } \alpha'$ |
| D‴ | $0 \quad q_1 \quad q_2' \quad k_- \quad q_c \quad k_+ \quad q_2 \quad q_3'$ | $0 \quad \tilde{\alpha} \quad \tilde{\delta} \quad \tilde{\epsilon} \quad \epsilon \quad \delta \quad \alpha \text{ or } \alpha'$ |
| E | $0 \quad k_- \quad q_c \quad q_2 \quad\quad k_+ \quad q_3'$ | $\tilde{\delta} \quad \tilde{\epsilon} \quad \epsilon \quad\quad \beta \quad \alpha \text{ or } \alpha'$ |
| E′ | $0 \quad q_1' \quad q_2' \quad k_- \quad q_c \quad k_+ \quad q_2 \quad q_3'$ | $\tilde{\delta} \quad \tilde{\alpha} \quad \tilde{\delta} \quad \tilde{\epsilon} \quad \epsilon \quad \delta \quad \alpha \text{ or } \alpha'$ |
| F | $0 \quad k_- \quad q_c \quad k_+ \quad\quad q_2 \quad q_3'$ | $\tilde{\delta} \quad \tilde{\epsilon} \quad \epsilon \quad\quad \delta \text{ or } \delta' \quad \alpha \text{ or } \alpha'$ |

Table 3: Same table as Table 2 but for $k > k_0$. See Table 5 for the meaning of the boundary values of $q$.

| Name | Boundaries | | | | | | | Configuration | | | | | |
|---|---|---|---|---|---|---|---|---|---|---|---|---|---|
| | | | | | | | | | | | | | |
| G | 0 | $q'_1$ | $k_-$ | $q'_2$ | $q_m$ | | | 0 | $\alpha$ | $\beta$ | $\gamma$ or $\gamma'$ | | |
| J | 0 | $q'_1$ | $k_-$ | $q'_2$ | $q'_3$ | $q''_3$ | $q_m$ | 0 | $\alpha$ | $\beta$ | $\gamma$ or $\gamma'$ | $\beta$ or $\beta'$ | $\gamma'$ |
| A | 0 | $q'_1$ | $k_-$ | $q'_2$ | $q'_3$ | $k_+$ | $q'_4$ | 0 | $\alpha$ | $\beta$ | $\gamma$ or $\gamma'$ | $\beta$ or $\beta'$ | $\alpha'$ |
| B | 0 | $q_1$ | $k_-$ | $q'_2$ | $q'_3$ | $k_+$ | $q'_4$ | $\delta$ | $\alpha$ | $\beta$ | $\gamma$ or $\gamma'$ | $\beta$ or $\beta'$ | $\alpha'$ |
| D | 0 | $k_-$ | $q_1$ | $q'_2$ | $q'_3$ | $k_+$ | $q'_4$ | $\delta$ | $\epsilon$ or $\epsilon'$ | $\beta$ | $\gamma$ or $\gamma'$ | $\beta$ or $\beta'$ | $\alpha'$ |
| E | 0 | $k_-$ | $q_1$ | $k_+$ | | | $q'_4$ | $\delta$ | $\epsilon$ or $\epsilon'$ | | | $\beta$ or $\beta'$ | $\alpha'$ |
| F | 0 | $k_-$ | $k_+$ | $q_1$ | | | $q'_4$ | $\delta$ | $\epsilon$ or $\epsilon'$ | | | $\delta'$ | $\alpha'$ |
| B' | 0 | $q'_1$ | $k_-$ | $k_+$ | $q'_4$ | | | 0 | $\alpha$ | $\beta$ or $\beta'$ | | | $\alpha'$ |
| D' | 0 | $q'_1$ | $k_-$ | $k_+$ | $q'_4$ | | | $\delta$ | $\alpha$ | $\beta$ or $\beta'$ | | | $\alpha'$ |
| H | 0 | $q'_1$ | $q''_1$ | | | | | 0 | $\alpha$ or $\alpha'$ | | | | |

Table with header: Superconfigurations of the $q$-integral for $k > k_0$

Table 4: Table of the boundary momenta (used in Table 2) in the region $k < k_0$. We use here the notation $\tilde{z}_k = z_k - 2$.

| Boundary momentum | Solution of | in the interval | condition of existence |
|---|---|---|---|
| $q_1$ | $\tilde{z}_k = \epsilon_{k+q}$ | $[0, k_0 - k]$ | $\epsilon_k > \tilde{z}_k > 1$ |
| $q_2$ | $\tilde{z}_k = \epsilon_{k+q}$ | $[k_0 - k, +\infty[$ | $\tilde{z}_k > 1$ |
| $q'_1$ | $\tilde{z}_k = \epsilon_{k-q}$ | $[0, k]$ | $\epsilon_0 > \tilde{z}_k > \epsilon_k$ |
| $q'_2$ | $\tilde{z}_k = \epsilon_{k-q}$ | $[k, k + k_0]$ | $\epsilon_0 > \tilde{z}_k > 1$ |
| $q'_3$ | $\tilde{z}_k = \epsilon_{k-q}$ | $[k + k_0, +\infty[$ | $\tilde{z}_k > 1$ |
| $q_c$ | $\epsilon_{k+q} = \epsilon_{k-q}$ | $[k - k_0, k + k_0]$ | $k < k_0$ |

Table 5: Table of the boundary momenta (used in Table 3) in the region $k > k_0$.

| Boundary momentum | Solution of | in the interval | condition of existence |
|---|---|---|---|
| For all $k$ | | | |
| $q_1$ | $z_k = \epsilon_{k+q} + \epsilon_c(q)$ | $[0, +\infty[$ | $\tilde{z}_k > \epsilon_k$ |
| $q_m$ | $z_k = 1 + \epsilon_c(q)$ | $[2k_0, +\infty[$ | $\tilde{z}_k > 1$ |
| $q'_4$ | $z_k = \epsilon_{k-q} + \epsilon_c(q)$ | $[k + k_0, +\infty[$ | $\tilde{z}_k > \omega_3$ |
| specific to $2k_0 > k > k_0$ | | | |
| $q'_1$ | $z_k = \epsilon_{k-q} + 2$ | $[0, k - k_0]$ | $\epsilon_k > \tilde{z}_k > 1$ |
| $q'_2$ | $z_k = \epsilon_{k-q} + 2$ | $[k - k_0, k]$ | $\epsilon_0 > \tilde{z}_k > 1$ |
| $q'_3$ | $z_k = \epsilon_{k-q} + \epsilon_c(q)$ | $[k, k + k_0]$ | $\epsilon_0 > \tilde{z}_k > \omega_2$ |
| $q''_3$ | $z_k = \epsilon_{k-q} + \epsilon_c(q)$ | $]q'_3, k + k_0]$ | $\omega_3 > \tilde{z}_k > \omega_2$ |
| specific to $3k_0 > k > 2k_0$ | | | |
| $q'_1$ | $z_k = \epsilon_{k-q} + 2$ | $[0, k - k_0]$ | $\epsilon_k > \tilde{z}_k > 1$ |
| $q'_2$ | $z_k = \epsilon_{k-q} + \epsilon_c(q)$ | $[k - k_0, k + k_0]$ | $\omega_3 > \tilde{z}_k > 1$ |
| specific to $k > 3k_0$ | | | |
| $q'_1$ | $z_k = \epsilon_{k-q} + \epsilon_c(q)$ | $[0, k - k_0]$ | $\epsilon_k > \tilde{z}_k > \omega'_2$ |
| $q''_1$ | $z_k = \epsilon_{k-q} + \epsilon_c(q)$ | $]q'_1, k - k_0]$ | $\omega'_3 > \tilde{z}_k > \omega'_2$ |
| $q'_2$ | $z_k = \epsilon_{k-q} + \epsilon_c(q)$ | $[k - k_0, k + k_0]$ | $\omega_3 > \tilde{z}_k > \omega'_3$ |

Table 6: Energy lines separating the resonance superconfigurations in Fig. 13.

| Energy line | Expression |
|---|---|
| $\omega_1$ | solution of $\epsilon_{k+q} = \epsilon_{k-q}$ for $q \in [0, k-k_0]$ |
| $\omega_2$ | $\epsilon_{k-q_{tg}+2\epsilon_{q_{tg}/2}}$ with $q_{tg}$ solution of $P_8(q_{tg}) = 0$ in $[k_0, 2k_0]$ (see Eq. (46)) |
| $\omega_2' = \epsilon_{th}^{1 \to 3} - 2$ | $3\epsilon_{k/3} - 2$ (lower edge of the continuum) |
| $\omega_3$ | $\epsilon_c(k+k_0) - 1$ |
| $\omega_3'$ | $\epsilon_c(k-k_0) - 1$ |
| $\omega_4$ | $\epsilon_{2k+k_0} + \epsilon_c(k+k_0) - 2$ |
| $\omega_4'$ | $\epsilon_{2k-k_0} + \epsilon_c(k-k_0) - 2$ |
| $\omega_4$ | $\epsilon_{2k+k_0} + \epsilon_c(k+k_0) - 2$ |

Table 7: Values of $k$ where energy lines cross.

| Name | Value in function of $k_0$ | Energy lines crossing |
|---|---|---|
| $k_0^-$ | | $\omega_1 = \epsilon_k = \epsilon_{2k-k_0} = 1$ |
| $k_1$ | $k_0/\sqrt{2}$ | $\omega_1 = \epsilon_0$ (avoided crossing, $\omega_1 \leq \epsilon_0$) |
| $k_2$ | $3k_0/5$ | $\epsilon_{2k-k_0} = \omega_1$ |
| $k_3$ | $k_0/2$ | $\epsilon_{2k-k_0} = \epsilon_0$ (avoided crossing, $\epsilon_{2k-k_0} \leq \epsilon_0$) |
| $k_4$ | $k_0/\sqrt{5}$ | $\epsilon_k = \omega_1$ |
| $k_5$ | $k_0/3$ | $\epsilon_{2k-k_0} = \epsilon_k$ |
| $k_6$ | $(\sqrt{2}-1)k_0/2$ | $\epsilon_{2k+k_0} = \epsilon_0$ |
| $k_7$ | $k_0/5$ | $\epsilon_{2k+k_0} = \epsilon_k$ |
| $k_0^+$ | | $\omega_2 = \omega_3 = \epsilon_k = \epsilon_{2k+k_0} = 1$ |
| $k_8$ | $(\sqrt{2}+1)k_0/2$ | $\epsilon_{2k+k_0} = \epsilon_0$ |
| $k_9$ | $\sqrt{2}k_0$ | $\epsilon_k = \epsilon_0$ |
| $k_{10}$ | $\sqrt{4k_0^2 + 2\sqrt{k_0^4 + 2\sqrt{1+k_0^4} - 2}} - k_0$ | $\omega_3 = \epsilon_0$ |
| $k_{11}$ | $2k_0$ | $\epsilon_0 = \omega_2$ ($\omega_2$ disappear beyond this point) |
| $k_{12}$ | $3k_0$ | Before this point $\omega_3'$ and $\omega_2'$ are identically equal to 1 |

and where $q_{tg}$ belongs to the interval $[k_0, 2k_0]$. This leads to the polynomial equation on $q_{tg}$:

$$16q_{tg}^8 - 32kq_{tg}^7 + 8\left(3k^2 - 5k_0^2\right)q_{tg}^6 - 8k\left(k^2 - 9k_0^2\right)q_{tg}^5 + \left(k^4 - 50k^2k_0^2 + 33k_0^4 + 21\right)q_{tg}^4$$
$$+ 4k\left(4k^2k_0^2 - 12k_0^4 - 9\right)q_{tg}^3 + \left[-2k^4k_0^2 + k^2\left(28k_0^4 + 25\right) - 10\left(k_0^6 + k_0^2\right)\right]q_{tg}^2$$
$$- 8k\left(k_0^4 + 1\right)(k-k_0)(k+k_0)q_{tg} + \left(k_0^4 + 1\right)\left(k^2 - k_0^2\right)^2 = 0. \quad (46)$$

## B  Angular integration in the self-energy

In this appendix, we derive analytic formulas of the angular integrals appearing in the self-eigenenergies $\Sigma^p$ and $\Sigma^{bc}$. In Eq. (18) the pair propagator is independent of the direction of $\mathbf{q}$ and goes out of the integral over $u = \mathbf{k} \cdot \mathbf{q}/kq$. The same is true for the spectral density of the pair propagator in Eq. (20). This leaves only elementary functions to integrate over $u$. We

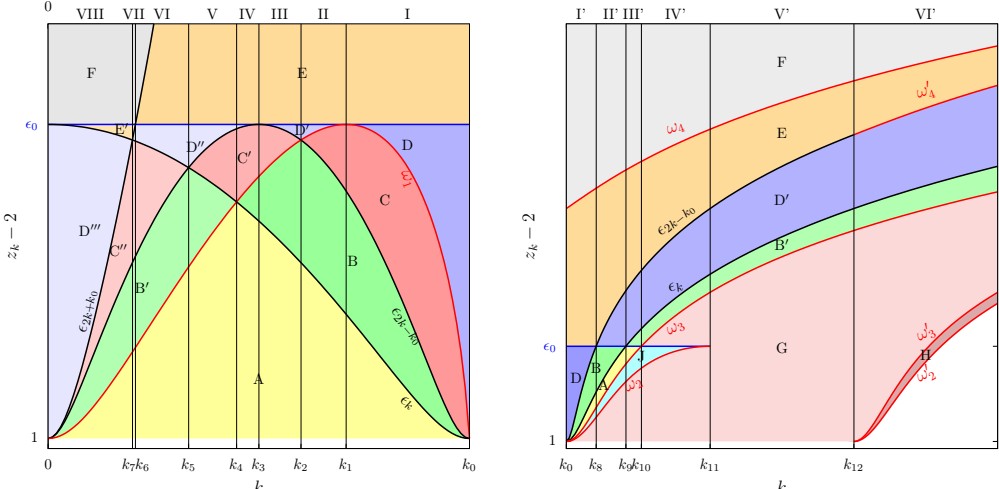

Figure 13: Diagram of the resonance superconfigurations for $k \leq k_0$ (left panel) and $k \geq k_0$ (right panel). The configurations denoted by Roman letters (see Tab. 2 for their signification) are shown in colors and delimited by energy lines shown in solid curves: in black $\epsilon_k$, $\epsilon_{2k-k_0}$ and $\epsilon_{2k+k_0}$, in blue $\epsilon_0 = \sqrt{k_0^2 + 1}$, in red the energies $\omega_i$ listed in Tab. 6 (note that $\omega_4$ and $\omega_4'$ continue $\epsilon_{2k+k_0}$ and $\epsilon_{2k-k_0}$ respectively at $k > k_0$ and $k > k_{12}$). The vertical lines delimit the $k$-intervals where the same succession of configurations is found when increasing $z_k$ from 3 to $+\infty$. To each sector between two vertical lines corresponds a diagram similar to Fig. 12.

thus introduce the integrals

$$I_{U^2|V^2}(z) = \int_{-1}^{1} \frac{du}{2}\left(1 \pm \frac{\xi_{\mathbf{k-q}}}{\epsilon_{\mathbf{k-q}}}\right)\frac{\Delta}{z + \epsilon_{\mathbf{k-q}}}, \tag{47}$$

$$I_{UV}(z) = \int_{-1}^{1} \frac{du}{2}\frac{\Delta}{\epsilon_{\mathbf{k-q}}}\frac{\Delta}{z + \epsilon_{\mathbf{k-q}}}, \tag{48}$$

and write the matrix elements of the self-energy in the form

$$\Sigma_{++}^{\mathrm{p}}(\mathbf{k}, z_k) = -\frac{1}{4\pi^2}\int_0^{+\infty}\frac{q^2 dq}{\partial_\omega \det \Gamma^{-1}(q, \omega_{\mathbf{q}})}\big[I_{V^2}(\omega_{\mathbf{q}} - z_k)\check{N}_{--}(q, \omega_{\mathbf{q}}) \tag{49}$$

$$-I_{U^2}(\omega_{\mathbf{q}} + z_k)\check{N}_{++}(q, \omega_{\mathbf{q}})\big], \tag{50}$$

$$\Sigma_{+-}^{\mathrm{p}}(\mathbf{k}, z_k) = \frac{1}{4\pi^2}\int_0^{+\infty}\frac{q^2 dq}{\partial_\omega \det \Gamma^{-1}(q, \omega_{\mathbf{q}})}\big[I_{UV}(\omega_{\mathbf{q}} - z_k) + I_{UV}(\omega_{\mathbf{q}} + z_k)\big]N_{+-}(q, \omega_{\mathbf{q}}), \tag{51}$$

$$\Sigma_{++}^{\mathrm{bc}}(\mathbf{k}, z_k) = -\frac{1}{4\pi^2}\int_0^{+\infty}q^2 dq\int_{\epsilon_c(q)}^{+\infty}d\omega_q\big[I_{V^2}(\omega_q - z_k)\rho_{++}(q, \omega_q) - I_{U^2}(\omega_q + z_k)\rho_{--}(q, \omega_q)\big],$$

$$\tag{52}$$

$$\Sigma_{+-}^{\mathrm{bc}}(\mathbf{k}, z_k) = -\frac{1}{4\pi^2}\int_0^{+\infty}q^2 dq\int_{\epsilon_c(q)}^{+\infty}d\omega_q\big[I_{UV}(\omega_q - z_k) + I_{UV}(\omega_q + z_k)\big]\rho_{+-}(q, \omega_q). \tag{53}$$

As before, we work here in units of $\Delta$, setting $\hbar^2 k^2/2m\Delta \to k^2$, $\hbar^2 q^2/2m\Delta \to q^2$ and

$$\frac{\xi_{\mathbf{k-q}}}{\Delta} = \xi_0 - 2kqu \to \xi_{\mathbf{k-q}}, \quad \text{with} \quad \xi_0 = k^2 + q^2 - \mu. \tag{54}$$

We use a Euler substitution to rationalize the integrand, setting

$$\epsilon_{\mathbf{k-q}} - \xi_{\mathbf{k-q}} = x \quad \Longleftrightarrow \quad u = \frac{x^2 + 2\xi_0 x - 1}{4kqx}, \tag{55}$$

$$du = \frac{x^2 + 1}{4kqx^2} dx. \tag{56}$$

Note in passing that this change of variable is monotonous (and increasing). In the new variable $x$, the integrals become

$$I_{U^2}(z) = \frac{1}{2kq} \int_{x_{\min}}^{x_{\max}} \frac{dx}{xP(x)}, \tag{57}$$

$$I_{V^2}(z) = \frac{1}{2kq} \int_{x_{\min}}^{x_{\max}} dx \frac{x}{P(x)}, \tag{58}$$

$$I_{UV}(z) = \frac{1}{2kq} \int_{x_{\min}}^{x_{\max}} \frac{dx}{P(x)}, \tag{59}$$

with the new integration boundaries

$$x_{\min} = \epsilon_{k+q} - \xi_{k+q}, \tag{60}$$
$$x_{\max} = \epsilon_{k-q} - \xi_{k-q}, \tag{61}$$

and the polynomial

$$P(x) = x^2 + 2zx + 1. \tag{62}$$

The roots of $P$ are

$$x_1 = -z + \sqrt{z^2 - 1} \underset{z = \omega + i0^+}{=} -\omega + \sqrt{\omega^2 - 1} + i\,\mathrm{sg}(\omega)0^+, \tag{63}$$

$$x_2 = -z - \sqrt{z^2 - 1} \underset{z = \omega + i0^+}{=} -\omega - \sqrt{\omega^2 - 1} - i\,\mathrm{sg}(\omega)0^+. \tag{64}$$

Effecting the partial fraction decomposition of the integrals, we finally get

$$I_{U^2}(\omega) = \frac{-I_1 - \omega I_2 + 2I_3}{4kq}, \tag{65}$$

$$I_{V^2}(\omega) = \frac{I_1 - \omega I_2}{4kq}, \tag{66}$$

$$I_{UV}(\omega) = \frac{I_2}{4kq}, \tag{67}$$

with

$$I_1 = \int_{x_{\min}}^{x_{\max}} dx \left( \frac{1}{x - x_1} + \frac{1}{x - x_2} \right) = \ln \left| \frac{(x_{\max} - x_1)(x_{\max} - x_2)}{(x_{\min} - x_1)(x_{\min} - x_2)} \right|$$
$$+ i\pi \mathrm{sg}(\omega) \left( \Theta(x_{\max} - x_1)\Theta(x_1 - x_{\min}) - \Theta(x_{\max} - x_2)\Theta(x_2 - x_{\min}) \right), \tag{68}$$

$$I_2 = \int_{x_{\min}}^{x_{\max}} \frac{\mathrm{d}x}{\sqrt{\omega^2 - 1}} \left( \frac{1}{x - x_1} - \frac{1}{x - x_2} \right) \tag{69}$$

$$= \begin{cases} \frac{1}{i\sqrt{1-\omega^2}} \ln \frac{(x_{\max}-x_1)(x_{\min}-x_2)}{(x_{\max}-x_2)(x_{\min}-x_1)} = \frac{2}{\sqrt{1-\omega^2}} \left( \arg(x_{\max}-x_1) - \arg(x_{\min}-x_1) \right), & \text{if} \quad 1 > |\omega|, \\ \frac{1}{\sqrt{\omega^2-1}} \Big[ \ln \left| \frac{(x_{\max}-x_1)(x_{\min}-x_2)}{(x_{\max}-x_2)(x_{\min}-x_1)} \right| \\ \quad + i\pi \mathrm{sg}(\omega) \left( \Theta(x_{\max}-x_1)\Theta(x_1-x_{\min}) + \Theta(x_{\max}-x_2)\Theta(x_2-x_{\min}) \right) \Big], & \text{if} \quad 1 < |\omega|, \end{cases}$$

$$I_3 = \ln \frac{x_{\max}}{x_{\min}}. \tag{70}$$

## C  UV contributions to $\Sigma^{\mathrm{bc}}$

In this appendix, we compute analytically the contributions of the high energies ($\omega_q \gg \epsilon_k, \Delta$) and short wavelengths ($q \gg k, k_F, \sqrt{2m\Delta}$) to the self-energy $\Sigma^{\mathrm{bc}}$. We consider two limits: either $\omega_q$ tends to $+\infty$ at fixed $q$, or $\omega_q$ and $q$ both tend to $+\infty$ with a fixed ratio $\omega_q/q^2$. Again, we use straightforward units of $\Delta$ (for instance $k_\Delta a \to a$ with $k_\Delta = \sqrt{2m\Delta}$), except for the matrix elements of the pair propagator, where we include a factor $(2\pi)^3$: $(2\pi)^3 \Delta N_{s,s'}/(k_\Delta^3 V) \to \check{N}_{s,s'}$ and correspondingly $(k_\Delta^3 V)\Gamma_{s,s'}/(2\pi)^3 \Delta \to \check{\Gamma}_{s,s'}$.

We first give the equivalent of the angular integrals computed in App. B. In the first limit, $\omega_q \gg q^2, k^2, |\mu|, 1$, one has

$$I_{U^2} \sim \frac{\frac{1}{x_{\min}} - \frac{1}{x_{\max}}}{4kq\omega_q}, \tag{71}$$

$$I_{V^2} \sim \frac{x_{\max} - x_{\min}}{4kq\omega_q}, \tag{72}$$

$$I_{UV} \sim \frac{\log \frac{x_{\max}}{x_{\min}}}{4kq\omega_q}, \tag{73}$$

and in the second limit, $\omega_q \approx q^2 \gg k^2, |\mu|, 1$:

$$I_{U^2} \sim \frac{2}{\omega_q + q^2}, \tag{74}$$

$$I_{V^2} \sim \frac{1}{2(\omega_q + q^2)q^4}, \tag{75}$$

$$I_{UV} \sim \frac{1}{(\omega_q + q^2)q^2}. \tag{76}$$

Next, we expand the pair propagator in the UV limit. Explicitly, the matrix elements of the bare propagator in the cartesian basis (as opposed to the phase-modulus basis [18, 51]) take the form:

$$\check{N}_{++}(z, \mathbf{q}) = \check{M}_{11}(z, \mathbf{q}) = -\frac{\pi^2}{a} + \int d^3k \left[ \frac{U_+^2 U_-^2}{z - \epsilon_+ - \epsilon_-} - \frac{V_+^2 V_-^2}{z + \epsilon_+ + \epsilon_-} + \frac{1}{2k^2} \right], \tag{77}$$

$$\check{N}_{--}(z, \mathbf{q}) = \check{M}_{22}(z, \mathbf{q}) = -\frac{\pi^2}{a} + \int d^3k \left[ \frac{V_+^2 V_-^2}{z - \epsilon_+ - \epsilon_-} - \frac{U_+^2 U_-^2}{z + \epsilon_+ + \epsilon_-} + \frac{1}{2k^2} \right], \tag{78}$$

$$\check{N}_{+-}(z, \mathbf{q}) = \check{M}_{12}(z, \mathbf{q}) = \int d^3k \left[ \frac{U_+ U_- V_+ V_-}{z + \epsilon_+ + \epsilon_-} - \frac{U_+ U_- V_+ V_-}{z - \epsilon_+ - \epsilon_-} \right], \tag{79}$$

with the notation $\epsilon_\pm = \epsilon_{\mathbf{k}\pm\mathbf{q}/2}$ and similarly for $U_\pm$ and $V_\pm$. Note that we gave in passing the equivalence between our notation $N_{s,s'}$ and the notation $M_{ij}$ of Refs. [18, 51]. At large $\omega_q$, those integrals are dominated by the large-$k$ region. We thus expand at $k^2 \gg 1, |\mu|/\Delta$ (but a priori $q \approx k$):

$$U_{\mathbf{k}} = 1 + O(k^{-4}), \tag{80}$$

$$V_{\mathbf{k}} = \frac{1}{2k^2} + O(k^{-4}), \tag{81}$$

$$\epsilon_{\mathbf{k}+\mathbf{q}/2} + \epsilon_{\mathbf{k}-\mathbf{q}/2} = 2k^2 + q^2/2 + O(1). \tag{82}$$

With this, we obtain the expansions of the matrix elements in the limit $\omega_q, q^2 \gg k^2, |\mu|, 1$:

$$\mathrm{Re}\check{N}_{--}(\omega_q, \mathbf{q}) = \pi^2 \left[ \frac{\sqrt{2\omega_q + q^2}}{2} - \frac{1}{a} \right] + O(1), \tag{83}$$

$$\mathrm{Im}\check{N}_{--}(\omega_q + i0^+, \mathbf{q}) = \frac{\pi^2}{q^7} n_{--}\left( \frac{2\omega_q}{q^2} \right) + O(q^{-8}), \tag{84}$$

$$\text{with} \quad n_{--}(\alpha) = -\frac{4\sqrt{\alpha - 1}}{(\alpha - 2)^2 \alpha^2} - \frac{\ln\left[ \frac{\sqrt{\alpha-1}+1}{\sqrt{\alpha-1}-1} \right]^2}{\alpha^3},$$

$$\check{N}_{++}(\omega_q + i0^+, \mathbf{q}) = -\pi^2 \left[ i\frac{\sqrt{2\omega_q - q^2}}{2} + \frac{1}{a} \right] + O(1), \tag{85}$$

$$\check{N}_{+-}(\omega_q + i0^+, \mathbf{q}) = \frac{\pi^2}{q^3} n_{+-}\left( \frac{2\omega_q}{q^2} \right) + O(q^{-4}), \tag{86}$$

$$\text{with } n_{+-}(\alpha) = \frac{4}{\pi} \mathcal{P} \int_0^{+\infty} \frac{t\,dt\,\mathrm{Argth}\left( \frac{2t}{t^2+1} \right)}{(1+t^2)^2 - \alpha^2} + \frac{i}{\alpha}\mathrm{Argth}\left( \frac{2\sqrt{\alpha - 1}}{\alpha} \right).$$

For the pair propagator $\Gamma$, this leads to

$$\mathrm{Re}\check{\Gamma}_{--}(\omega_q, \mathbf{q}) = \frac{1}{\pi^2 \left[ \frac{\sqrt{2\omega_q + q^2}}{2} - \frac{1}{a} \right]} + O(q^{-3}), \tag{87}$$

$$\mathrm{Im}\check{\Gamma}_{--}(\omega_q + i0^+, \mathbf{q}) = \frac{1}{\pi^2 q^9} \gamma_{22}\left( 2\omega_q/q^2 \right) + O(q^{-10}), \tag{88}$$

$$\text{with} \quad \gamma_{22}(\alpha) = -\frac{4}{\alpha + 1}\left( n_{--}(\alpha) - \frac{2\mathrm{Re}\,n_{+-}^2(\alpha)}{\sqrt{\alpha - 1}} \right),$$

$$\check{\Gamma}_{++}(\omega_q + i0^+, \mathbf{q}) = -\frac{1}{\pi^2 \left[ i\frac{\sqrt{2\omega_q - q^2}}{2} + \frac{1}{a} \right]} + O(q^{-2}), \tag{89}$$

$$\check{\Gamma}_{+-}(\omega_q + i0^+, \mathbf{q}) = \frac{4}{\pi^2 q^5} \frac{n_{+-}(2\omega_q/q^2)}{\sqrt{4\omega_q^2/q^4 - 1}} + O(q^{-6}). \tag{90}$$

In the more stringent limiting case $\omega_q \gg q^2$, those asymptotic behaviors simplify to

$$\mathrm{Re}\check{N}_{--} = \frac{\pi^2 \sqrt{\omega_q}}{\sqrt{2}} + O(1), \tag{91}$$

$$\mathrm{Im}\check{N}_{--} = -\frac{\pi^2}{\sqrt{2}\omega_q^{7/2}} + O(\omega_q^{-4}), \tag{92}$$

$$\mathrm{Re}\check{N}_{++} = -\frac{\pi^2}{a} + O(\omega_q^{-1/2}), \tag{93}$$

$$\mathrm{Im}\check{N}_{++} = -\frac{\pi^2 \sqrt{\omega_q}}{\sqrt{2}} + O(1), \tag{94}$$

$$\check{N}_{+-} = \frac{\pi^2}{\sqrt{2}\omega_q^{3/2}}(i-1) + O(\omega_q^{-2}), \tag{95}$$

$$\mathrm{Re}\check{\Gamma}_{--} = \frac{\sqrt{2}}{\pi^2 \sqrt{\omega_q}} + O(\omega_q^{-1}), \tag{96}$$

$$\mathrm{Im}\check{\Gamma}_{--} = \frac{\sqrt{2}}{\pi^2 \omega_q^{9/2}} + O(\omega_q^{-5}), \tag{97}$$

$$\mathrm{Re}\check{\Gamma}_{++} = -\frac{2}{\pi^2 a \omega_q} + O(\omega_q^{-3/2}), \tag{98}$$

$$\mathrm{Im}\check{\Gamma}_{++} = \frac{\sqrt{2}}{\pi^2 \sqrt{\omega_q}} + O(\omega_q^{-1}), \tag{99}$$

$$\Gamma_{12} = \frac{\sqrt{2}}{\pi^2 \omega_q^{5/2}}(1+i) + O(\omega_q^{-3}). \tag{100}$$

We then extract the UV behavior of the integrand of Eqs. (52)–(53), first at finite $q$:

$$-\check{\rho}_{++}(q, \omega_q) I_{V^2}(\omega_q \pm z_k) \sim \frac{\sqrt{2}(x_{\max} - x_{\min})}{4\pi^3 kq \omega_q^{3/2}}, \tag{101}$$

$$-\check{\rho}_{--}(q, \omega_q) I_{U^2}(\omega_q \pm z_k) \sim \frac{\sqrt{2}(x_{\min}^{-1} - x_{\max}^{-1})}{4\pi^3 kq \omega_q^{11/2}}, \tag{102}$$

$$-\check{\rho}_{+-}(q, \omega_q) I_{UV}(\omega_q \pm z_k) \sim \frac{\sqrt{2}(\log(x_{\max}/x_{\min}))}{4\pi^3 kq \omega_q^{7/2}}. \tag{103}$$

With this, we compute analytically the integral over $\omega_q$ from a UV cutoff $\Omega_0 \gg 1$ to $+\infty$:

$$J_{++}(q) \equiv -\int_{\Omega_0}^{+\infty} \check{\rho}_{++}(q, \omega_q) I_{V^2}(\omega_q \pm z_k) \underset{\Omega_0 \to +\infty}{\sim} \frac{\sqrt{2}(x_{\max} - x_{\min})}{2\pi^3 kq \Omega_0^{1/2}}, \tag{104}$$

$$J_{--}(q) \equiv -\int_{\Omega_0}^{+\infty} \check{\rho}_{--}(q, \omega_q) I_{U^2}(\omega_q \pm z_k) \underset{\Omega_0 \to +\infty}{\sim} \frac{\sqrt{2}(x_{\min}^{-1} - x_{\max}^{-1})}{18\pi^3 kq \Omega_0^{9/2}}, \tag{105}$$

$$J_{+-}(q) \equiv -\int_{\Omega_0}^{+\infty} \check{\rho}_{+-}(q, \omega_q) I_{UV}(\omega_q \pm z_k) \underset{\Omega_0 \to +\infty}{\sim} \frac{\sqrt{2}(\log(x_{\max}/x_{\min}))}{10\pi^3 kq \Omega_0^{5/2}}. \tag{106}$$

Finally we compute analytically the integral over $q$ from $Q \gg 1$ to $+\infty$. At such large $q$, the subintegral over $\omega_q$ runs from $q^2/2$ ($a = 1$) to infinity. Setting $\Omega_0 = q^2/2$ in Eqs. (104–106),

and expanding for $q \to +\infty$, we obtain:

$$q^2 J_{++}(q) \underset{q \to +\infty}{\sim} \frac{2}{\sqrt{3}\pi^3 q^5}, \quad \text{such that} \quad \int_Q^{+\infty} q^2 J_{++}(q) = \frac{1}{2\sqrt{3}\pi^3 Q^4}, \quad (107)$$

$$\int_Q^{+\infty} q^2 J_{--}(q) = O(1/Q^8), \quad (108)$$

$$\int_Q^{+\infty} q^2 J_{+-}(q) = O(1/Q^6). \quad (109)$$

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
