# Peer review of "Quasiparticle disintegration in fermionic superfluids"

_SciPost Physics, doi:SciPost Phys. 12, 108 (2022)_

## Round 1 · Referee Report · Anonymous (Referee 1) · 2021-12-17

Strengths

The paper analyses for the first time various decay modes of quasiparticles in a superfluid Fermi gas, with very important implications for experiments in the near future.

The paper is extremely clear. The results are presented in a very detailed and understandable way.

The decays the authors analyze have an extremely involved phenomenology, also modifying the quasiparticle spectrum: the paper clearly advances the knowledge of fermionic superfluids.

Weaknesses

A possible (actually quite natural) extension of the theory to superconductors is only hinted at, at the very end of the paper. A more detailed analysis would increase the relevance of the paper.

Report

In the paper "Quasiparticle disintegration in fermionic superfluids" S. Van Loon and coauthors analyze the quasiparticle spectrum of a superfluid Fermi gas, identifying various quasiparticle disintegration processes.

The authors analyzed other (bosonic) decay types in a past paper, and here they shift the focus to fermionic decays. However, one cannot define this work incremental, since they develop new techniques, and the decays they analyze are extremely interesting on their own, while also presenting a complex phenomenology that the authors analyze in great detail.

Experimental connections and verifications are properly mentioned, however it could be useful for the reader, if the authors commented a bit more about this in the Conclusions. For instance: the quasiparticle spectrum is heavily distorted near the $1 \to 3$ disintegration threshold. Is this accessible to experiments?

The paper is carefully written and edited, and is generally very easy to follow.

On the "Requested changes" section I comment a bit about one single point which is not immediately clear to me, about the approximation scheme used. It would be very useful if the authors would briefly clarify this. There I also ask for clarifications about other (possible) decay processes.

In summary, I believe the paper could meet the SciPost Physics acceptance criteria after a minor revision, since the central result is clearly a groundbreaking theoretical discovery.

Requested changes

I would ask the authors to clarify the following point: which scheme is used to calculate the chemical potential and the pairing gap \Delta? Are they calculated at mean-field level or taking in account Gaussian fluctuations? Either way, what's the best scheme to use in conjunction with the ladder resummation scheme?

Slightly related: the authors derive some results for the far-off-resonant $g \to 0$ case, while also presenting some other results for the unitary case, where clearly the assumption above doesn't hold, and there one also needs a beyond-mean-field equation of state to obtain quantitatively-correct results. That's the only point of the paper that I found a bit confusing. I think the paper would greatly benefit if the scope of applicability of each result derived was marked with greater emphasis.

It would be very interesting to briefly mention higher-order decay processes, and maybe even (is it possible?) estimate their order of magnitude. Of course a full calculation is most likely totally unfeasible!

Similarly, a more detailed connection to superconductors would make the paper relevant for a broader audience. This is an optional suggestion.

  • validity: top
  • significance: high
  • originality: top
  • clarity: top
  • formatting: excellent
  • grammar: excellent

Author:  Senne Van Loon  on 2022-01-20  [id 2115]

(in reply to Report 1 on 2021-12-17)

Dear Referee,

We thank you for revising our manuscript and your suggestions. We have made changes to clarify the points you mentioned:

We now mention the scheme we use to calculate the chemical potential and the order parameter. Within the ladder resummation scheme, the number equation could be altered to include Gaussian fluctuations, which would be the most quantitative. However, here our main results are shown by varying $\mu/\Delta$, to avoid making a choice of the approximation of the number equation. Whenever we do relate $\mu/\Delta$ to the interaction $1/k_F a$, we use the mean-field equations.

The results we derive in the $g\to 0$ case are only used to clarify which processes contribute to the self-energy and to write down analytic forms for the coupling amplitudes. In all of our numerical analysis, we use the general form of the self-energy valid for all coupling. We have made this more clear in the main text.

It is, like you say, very difficult to fully include higher order decay processes and even estimate their order of magnitude. However, it is possible to imagine what these processes would be. We have added a paragraph describing these processes, and argue why they should give a small correction with respect to the processes we already include.

Concerning superconductors, we fear the generalization of our study may not be as straightforward as suggested by the Referee. Superconducting electrons are not as well isolated from their environment as ultracold fermions. As a consequence, they have, besides their intrinsic lifetime due to the kind of processes we consider here, extrinsic damping mechanisms, such as impurity scattering and emission of lattice phonons. Following your remark, we have refined our discussion of the case of superconductors in conclusion.

Finally, we have clarified that our predictions could be verified by experiments using momentum-resolved rf-spectroscopy. With this technique, also the heavily distorted spectrum near the $1\to3$ threshold should be visible.

Yours sincerely,

The authors

---

## Round 1 · Referee Report · Anonymous (Referee 2) · 2021-12-29

Report

The manuscript provides an in-depth analysis of the single-particle fermionic excitation spectrum in a superfluid Fermi gas at zero temperature. The calculations are based on a t-matrix self-energy approach extended to the superfluid phase. The paper focuses in particular on a decay channel which can be interpreted as the decay of a single quasi-particle into three quasi-particles, complementing a previous work by the same authors which focused instead on the decay channel associated with the emission of a bosonic excitation (Bogoliubov sound mode) by a quasi-particle.

The paper is quite technical in nature, but I think it could be of interest to readers specialized in the theory of superfluid Fermi gases and the BCS-BEC crossover. The paper is clearly written and the appendices provides all the details of the analytic calculations performed by the authors. Even though I am not fully convinced that the paper is really 'groundbreaking' or a 'breakthrough', as expected by SciPost, I regard it as a valuable contribution and I am therefore inclined to support its publication on SciPost Physics, provided the following points are addressed satisfactorily by the authors.

1) The authors did not explain anywhere which kind of particle number equation they have used to relate the chemical potential \mu and order parameter \Delta to the density, and hence to the dimensionless coupling 1/(k_F a). This is especially relevant to Figs. 9 and 10, but also whenever the ratio \mu/Delta is associated to a given coupling, like in Figs. 5 and 6. The authors should thus explain the kind of number equation they have used, and provide some details of its numerical solution.
2) From the reported ratio \mu/Delta = 0.86 at unitarity, I deduce that the authors have used the BCS mean-field equation for the order parameter \Delta. The authors should mention it explicitly somewhere in the paper.
3) In fig.10 the authors compare the excitation gap \epsilon* with the order parameter \Delta as a function of the coupling 1/(k_F a). In the paragraph (starting on pag. 24 and finishing on pag. 25) the authors comment that the ratio epsilon*/Delta tends to one in the BCS limit and regard it as a serious limitation of their approach, which does not recover the reduction by a factor 2.2 of the order parameter predicted long ago by Gorkov and Melik-Barkhudarov for a contact potential in the weak-coupling limit. I do not understand this comment. The reduction by by Gorkov and Melik-Barkhudarov concerns the order parameter Delta. It consists in a reduction of Delta in comparison with its BCS mean-field value in units of E_F, for a given coupling 1/(k_F a). To obtain such a reduction, one needs to modify the equation for the order parameter (in order to take into account an effective screening of the attractive interaction due to particle-hole excitations in the medium). There is no reason upon which one should expect such a reduction to appear in the ratio between the excitation gap epsilon* and the order parameter Delta. In the BCS limit, and for an ‘exact’ theory, I would rather expect this ratio to tend to one, with both Delta and epsilon* tending to the BCS value reduced by the factor 2.2. I think that the authors should argue better on this point, and/or modify accordingly this part of their manuscript.

Requested changes

The above points 1-3 should be addressed satisfactorily in a revision of the manuscript.

  • validity: high
  • significance: high
  • originality: high
  • clarity: high
  • formatting: excellent
  • grammar: perfect

Author:  Senne Van Loon  on 2022-01-20  [id 2116]

(in reply to Report 2 on 2021-12-29)

Dear Referee,

We thank you for revising our manuscript and your suggestions. We have made changes to clarify the points you mentioned:

1) and 2) We have added a discussion on the equation of state and how we relate the chemical potential and the order parameter to the coupling $1/k_F a$. As much as possible, we present our main results as a function of $\mu/\Delta$ to avoid using a number equation. Whenever we relate this parameter to $1/k_F a$, we use the mean-field number equation, which has an analytic solution when solved together with the mean-field equation for the order parameter. We now explicitly mention both equations in the main text.

3) We have clarified our discussion on the Gorkov and Melik-Barkhudarov (GMB) correction. Since we do not refine the mean-field equation for the order parameter, our value of $\Delta$ does not include the GMB correction (our $\Delta$ is $\Delta_{\rm BCS}$). The fact that we find that $\epsilon^\ast/\Delta_{\rm BCS}\to1$ thus implies that the energy gap $\epsilon^\ast$ coincides (in the weak-coupling regime) with the BCS value of the order parameter $\Delta_{\rm BCS}$ and not with the corrected value $\Delta_{\rm GMB}$. We see this as a limitation of our theory, although, as you mentioned, whether the GMB correction applies also to the quasiparticle gap (and more generally whether the gap and order parameter remain equal beyond BCS approximation) remains an open question. In view of the latest experimental results (Ref.~[38]), which find a reduction of the gap compatible with the GMB prediction (for the order parameter), it is indeed a reasonable conjecture that the ratio between the two corrected values tend to one. To show this, a consistent theory, where both the equation for the order parameter and the quasiparticle self-energy are modified, is required, and it should include more diagrams than the $t$-matrix contribution considered here.

Yours sincerely,

The authors

---

## Round 2 · Referee Report · Anonymous (Referee 1) · 2022-1-20

Report

The authors modified the manuscript, replying in a very satisfactory way to my remarks in my original referee report. It is therefore my pleasure to recommend the manuscript for publication in SciPost.

---

## Round 2 · Referee Report · Anonymous (Referee 2) · 2022-1-21

Report

The authors have addressed satisfactorily my remarks; I now fully support publication of the present manuscript on SciPost.

---

## Round 2 · List of Changes

*Added a discussion on the equation of state and how to relate $\mu/\Delta$ to the coupling $1/k_F a$.
*Added a discussion on possible higher-order processes contributing to the quasiparticle spectrum .
*Clarified that all numerical results make use of the full self-energy, valid for all couplings.
*Clarified why we expect to find a reduction of the corrected gap in the BCS limit, and how to include corrections similar to those predicted by Gor'kov and Melik-Barkhudarov.
*Rewrote the conclusion to reflect the fact that our predictions are experimentally verifiable, and to clarify the connection with superconductors.

---

## Editorial Decision

published